

# Interannual variability of the boreal summer tropical UTLS in observations and CCMVal-2 simulations

Markus Kunze[1], Peter Braesicke[2], Ulrike Langematz[1], and Gabriele Stiller[2]

[1]Institut für Meteorologie, Freie Universität Berlin, Carl-Heinrich-Becker-Weg 6–10, 12165 Berlin, Germany
[2]Institut für Meteorologie und Klimaforschung, Karlsruher Institut für Technologie, H.-v.-Helmholtz-Platz 1, 76344 Leopoldshafen, Germany.

*Correspondence to:* Markus Kunze (markus.kunze@met.fu-berlin.de)

**Abstract.** During boreal summer the upper troposphere/lower stratosphere (UTLS) in the northern hemisphere shows a distinct maximum in water vapour ($H_2O$) mixing ratios and a coincident minimum in ozone ($O_3$) mixing ratios, both confined within the Asian monsoon anticyclone (AMA). This well known feature has been related to transport processes emerging above the convective systems during the Asian summer monsoon (ASM), further modified by the dynamics of the AMA. We assess

the ability of chemistry climate models (CCMs) to reproduce the climatological characteristics and variability of $H_2O$, $O_3$ and temperature in the UTLS during the boreal summer in comparison to MIPAS satellite observations and ERA-Interim re–analyses. By using a multiple linear regression model the main driving factors, the strength of the ASM, the quasi-biennial oscillation (QBO), and the El Niño-Southern Oscillation (ENSO), are separated. The results of the regression analysis show for ERA-Interim and the CCMs enhanced $H_2O$ and reduced $O_3$ mixing ratios within the AMA for stronger ASM seasons. The

CCM results can further confirm earlier studies which emphasize the importance of the Tibetan Plateau/southern slope of the Himalayas as the main source region for $H_2O$ in the AMA. The results suggest that $H_2O$ is transported towards higher latitudes at the north–eastern edge of the AMA, rather than transported towards low equatorial latitudes to be fed into the tropical pipe. The regression patterns related to ENSO show a coherent signal for temperatures and $H_2O$ mixing ratios for ERA-Interim and the CCMs, and suggest a weakening of the ASM during ENSO warm events. The QBO modulation of the lower stratospheric

temperature near the Equator is well represented in the CCMs. Its influence on $H_2O$ and $O_3$ mixing ratios is consistent but weaker.

## 1 Introduction

The future evolution of the abundances of chemically active trace gases has a major influence on the development of strato-spheric ozone ($O_3$) concentrations. In addition to the chemical composition of the stratosphere, the trace gases also influence

the radiation budget of the atmosphere. To identify the major transport processes responsible for troposphere–stratosphere transport (TST), is therefore an important issue.

The transport of tracers from the troposphere to the stratosphere is largely realized through the tropical tropopause layer (TTL) (Fueglistaler et al., 2009, and references therein). It consists of a rapid rise by convection to the lower boundary of the TTL, slow ascent forced by radiative heating in the TTL and the wave driven further ascent within the Brewer-Dobson circu-





lation (BDC). The amount of water vapour ($H_2O$) that enters the stratosphere depends on the lowest temperature encountered during ascent, where moist air is freeze dried until the $H_2O$ mixing ratios are as low as the saturation mixing ratio of the region passed through (Brewer, 1949). In addition to the slow vertical transport, that takes place in a fast horizontally directed flow (Holton and Gettelman, 2001), there is evidence for direct injections of $H_2O$ into the stratosphere by convection overshooting

the tropopause (e.g. Sherwood and Dessler, 2000), or direct injections of ice particles into the lower stratosphere, that sublimate and moisten the lower stratosphere (Corti et al., 2008).

In recent years the off–equatorial Asian summer monsoon (ASM) has also been emphasized to contribute to the transport of trace gases from the troposphere to the stratosphere (Gettelman et al., 2004; Bannister et al., 2004; Fu et al., 2006; Wright et al., 2011; Randel et al., 2010, 2015). Observational and model studies (e.g. Rosenlof et al., 1997; Pan et al., 1997; Gettelman

et al., 2004; Randel and Park, 2006; Park et al., 2007; Kunze et al., 2010; Ploeger et al., 2013) have shown that during boreal summer the maximum in $H_2O$ coincides with a minimum in $O_3$, confined to the Asian monsoon anticyclone (AMA) in the upper troposphere/ lower stratosphere (UTLS). The AMA can be explained as a dynamic response to diabatic heating by the underlying convective activity (Gill, 1980). It has been shown that the area of main convective activity, identified by low values of outgoing longwave radiation (OLR) in the Bay of Bengal (BoB) and its surroundings, is displaced to the southeast of

the AMA and the $H_2O$ maximum at 100 hPa (e.g. Park et al., 2007). As analysed by Park et al. (2009) the convective systems transport tracers from the source region up to ∼200 hPa. At that altitude, near the level of main convective outflow, the divergent flow further advects the tracers mainly to the south–west and to the north–east towards the North Pacific Ocean. These outflows in the longitudinal direction have been classified as transverse circulations by Yang et al. (1992) and Webster et al. (1998), with the outflow to the north–east identified as part of the Walker circulation, and the outflow in the latitudinal direction classified as

lateral circulation which is part of the reversed Hadley circulation. In contrast, the role of the orography of the Tibetan Plateau (TP) and heating above the TP in forming the AMA has been studied by Liu et al. (2007) with a simplified general circulation model (GCM). They found heating above the TP to be the predominant forcing of the upper level anticyclonic flow.

In order to analyse the origins and the transport pathways of constituents in the UTLS of the ASM region backward trajectories studies have widely been used (e.g. Jensen and Pfister, 2004; Fueglistaler et al., 2004; James et al., 2008; Kremser

et al., 2009; Ploeger et al., 2011; Wright et al., 2011; Bergman et al., 2013). To estimate the relative role of specific regions within the ASM area, four main source regions are usually compared: the BoB and the Indian subcontinent (IND) (both regions sometimes combined as MON), the southern slope of the Himalayas (SS), and the TP. Fu et al. (2006) and Wright et al. (2011) identified the regions of the TP and the SS as most important to bypass the lowest cold point and moisten the air within the AMA. This is consistent with Heath and Fuelberg (2014), who used a model system that explicitly resolved convection to show

that 90% of the air parcels influenced by convection within the AMA are connected to the convection over the TP and the SS. They emphasised that, due to the high elevation of the TP, convection does not necessarily have to be particularly strong to reach the AMA. In contrast, Chen et al. (2012) identified the TP and the SS to be only of minor importance as source region for tracers in the tropopause layer, and highlighted that the region extending from the western Pacific to the South China Seas is most important.




The extra tropical lower stratosphere exhibits a strong seasonal cycle in $H_2O$ mixing ratios, which Ploeger et al. (2013) argued to be almost entirely created by horizontal transport on isentropic levels from low latitudes. They show that filaments of high $H_2O$ mixing ratios at 390 K, drawn out of the ASM region on the eastern side of the AMA, are responsible for $H_2O$ transport from low to high latitudes during boreal summer. The potential of this kind of $H_2O$ transport out of the ASM region for

moistening the extra tropical lower stratosphere was already investigated by Dethof et al. (1999). Whereas there is agreement about the ASM in moistening the lower stratosphere at higher latitudes, the discussion of the role of the ASM contributing to the moist phase of the stratospheric tropical $H_2O$ tape–recorder signal is controversial. E.g., Wright et al. (2011) found only a minor contribution of the ASM to the mean tropical stratospheric $H_2O$, while others studies (e.g. Gettelman et al., 2004; Bannister et al., 2004) highlighted the large impact of the ASM on the moist phase of the tropical $H_2O$ tape–recorder. Randel

et al. (2010) argued for a direct link between the pollutants produced in the East–Asian region and enhanced hydrogen cyanid (HCN) mixing ratios in the tropical lower stratosphere transported upward through the upper tropospheric AMA.

Superimposed on the climatological $H_2O$, and $O_3$ concentrations in the UTLS, described so far, is an interannual variability caused by internal modes of variability like El Niño Southern Oscillation (ENSO), and the Quasi–Biennial Oscillation (QBO), or the ASM. Additional variability arises from external forcing, like the 11-year solar cycle, or from sporadic events like

volcanic eruptions. These components are not independent of each other, as for example the ASM itself is influenced by ENSO (e.g. Webster and Yang, 1992), the QBO (Giorgetta et al., 1999) or the 11-year solar cycle (van Loon and Meehl, 2012). As shown by Kunze et al. (2010), the strength of the ASM has some influence on the observed $H_2O$ maximum and $O_3$ minimum mixing ratios confined by the AMA, with increasing $H_2O$ and decreasing $O_3$ mixing ratios during strong ASM seasons. However, a recent study by Randel et al. (2015), suggested that increasing $H_2O$ mixing ratios within the AMA can

also be related to weaker ASM seasons.

The aim of this study is to assess, by a comparison to satellite data, chemistry climate model (CCM) simulations of the recent past with respect to their ability to capture the $H_2O$ and $O_3$ climatological distribution in the UTLS during the ASM. In addition, the ASM related mean circulation and temperature patterns will be compared with a re–analysis dataset. Further we want to identify the relative importance of the ASM, in comparison to ENSO and the QBO, in modulating the $H_2O$ and $O_3$ mixing

ratios in the UTLS. The paper is organised as follows: After introducing the data (Section 2) and methods (Section 3) aspects of the climatological state of the CCMs in the UTLS are compared with observations and re–analyses during July/August (JA) in Section 4. The interannual variability of the temperature and the $H_2O$ and $O_3$ concentrations is then investigated in Section 5, followed by a summary and the conclusions in Section 6.

## 2   Models and Data

We use data from CCMs collected for Phase II of the Chemistry-Climate-Model validation activity (CCMVal-II) for Stratospheric Processes and their Role in Climate (SPARC). We focus on the so-called REF-B1 simulations of the recent past covering the period from 1960–2004. The SPARC Report No 5 on Chemistry-Climate-Model validation (SPARC CCMVal, 2010) gives





**Table 1.** Main characteristics and specifications of the Chemistry-Climate models used. More comprehensive information can be found in Morgenstern et al. (2010).

| CCM | Horiz. Res. | Levels/ Upp. Bound. | Levels: 300–100 hPa | QBO |
|---|---|---|---|---|
| CCSRNIES | T42 | 34 / 0.012 hPa | 6 | nudged |
| CMAM | T31 | 71 / 0.00081 hPa | 7 | no |
| CNRM-ACM | T42/T21 | 60 / 0.07 hPa | 8 | no |
| E39CA | T30 | 39 / 10 hPa | 15 | nudged |
| EMAC | T42 | 90 / 0.01 hPa | 12 | weakly nudged |
| EMAC-FUB | T42 | 39 / 0.01 hPa | 5 | nudged |
| GEOSCCM | 2° x 2.5° | 72 / 0.015 hPa | 7 | no |
| SOCOL | T30 | 39 / 0.01 hPa | 5 | nudged |
| UMUKCA-UCAM | 2.5° x 3.75° | 60 / 84 km | 7 | internal |
| WACCM | 1.9° x 2.5° | 66 / 5.96x10$^{-6}$ hPa | 7 | nudged |

a comprehensive overview of the details of the CCMs used in this study, therefore only the main features are summarized in section 2.1.

## 2.1 Chemistry Climate Model data

Here, we use monthly mean data of temperature, zonal and meridional wind, vertical velocity, $H_2O$, $O_3$, longwave and short-wave heating rates, varying in longitude, latitude, pressure, and time. Only a subset of the CCMs, participating in the CCMVal-II activity have provided all required data to the CCMVal archive which limits the analyses to the CCMs listed in Table 1. Most CCMs have their upper boundary in the upper mesosphere or lower thermosphere, E39CA is the only model with an upper boundary in the middle stratosphere at 10 hPa. The vertical resolution in the UTLS region (300–100 hPa) ranges from 5 (EMAC-FUB and SOCOL) to 15 (E39CA) levels.

## 2.1.1 Model runs

The specifications of the CCMVal REF-B1 scenario were designed to produce best estimate model simulations of the recent past from 1960–2006 (Eyring et al., 2008). They define a transient setup that includes all anthropogenic and natural forcings, with greenhouse gases (GHGs) according to IPCC (2001) (updated with NOAA observations to 2006), ozone depleting substances (ODSs) according to WMO (2007), prescribed monthly varying sea surface temperatures (SSTs) and sea ice concentrations (SICs) from the global HadISST1 data set (Rayner et al., 2006). To account for the effect of the major volcanic eruptions on the temperatures in the stratosphere and troposphere, additional heating rates for the stratosphere and cooling of the surface have been prescribed or calculated from an aerosol data set, where possible. The effect of volcanic aerosol on heterogeneous chemistry is taken into account by prescribing a surface area density data set. The solar variability of the 11–year solar cycle





and the 27–day solar rotational period is included in some simulations by spectrally resolved solar irradiances on a daily basis (Lean et al., 2005). The quasi–biennial oscillation (QBO) is not included in all CCMs (see Table 1). In a subset of CCMs it is nudged, or it develops internally (UMUKCA-UCAM, EMAC) in CCMs with sufficiently high vertical resolution and an adequate gravity wave parametrization. EMAC has an internally generated QBO and weak nudging is applied to force the model towards the observed timing of the QBO phase.

## 2.2  Re-analyses and satellite data

The European Centre for Medium–Range Weather Forecasts (ECMWF) interim re-analyses (ERA-Interim) data from 1979–2014 (Dee et al., 2011) are used in this study to assess the monsoonal wind structure, the velocity potential, and the stream-function. The ERA-Interim water vapor and ozone data are used in regression analyses, when longer time series covering a large part of the modelled time period are necessary. The period used for the ERA-Interim data does not exactly match the period of the REF-B1 simulations of the CCMs, but due to the overlapping period from 1979–2004, covering nearly 60% of the REF-B1 period, a comparison with ERA-Interim is still useful. The quality of the ERA-Interim ozone data has been assessed by Dragani (2011), showing a better quality compared to the previous ERA-40 re-analysis (Uppala et al., 2005).

As observational reference for the climatological JA water vapor and ozone mixing ratios on the 380 and 370 K isentropic level we use the Michelson Interferometer for Passive Atmospheric Sounding (MIPAS) satellite data of $H_2O$ (Milz et al., 2009; von Clarmann et al., 2009) and $O_3$ (Steck et al., 2007; von Clarmann et al., 2009). MIPAS measures $H_2O$ and $O_3$, among numerous other species, as a limb emission midinfrared sounder with high spectral resolution from a sun-synchronous polar orbit at about 800 km altitude. It covers the atmosphere from cloud top to 70 km by scanning from top to bottom with a step width of 1.5 km (UTLS, since 2005) to 8 km (mesosphere, before 2005). Data are recorded every 400 km along the orbit, with 14.4 orbits per day, providing one profile per day roughly every 4° latitude and 12.5° longitude. Cloud contamination reduces the achievable coverage.

The atmospheric distributions of $H_2O$ and $O_3$ used in this study were derived using the MIPAS level-2 data processor at the Institut für Meteorologie und Klimaforschung–Instituto de Astrofísica de Andalucía (von Clarmann et al., 2003) from observations of 57 days overall during July and August 2003, and 2005–2009 (6 years). The precision, accuracy, and vertical resolution of single profiles in the relevant altitude range of $H_2O$ ($O_3$) is 5–6%, 8–17%, and 2.3–3.3 km (3.8–12.6%, 9.6–17.0%, 2.4–2.9 km), respectively (von Clarmann et al., 2009).

NOAA interpolated monthly average outgoing longwave radiation (OLR) (Liebmann and Smith, 1996) from 1975–2013 are used as a proxy for deep convection.





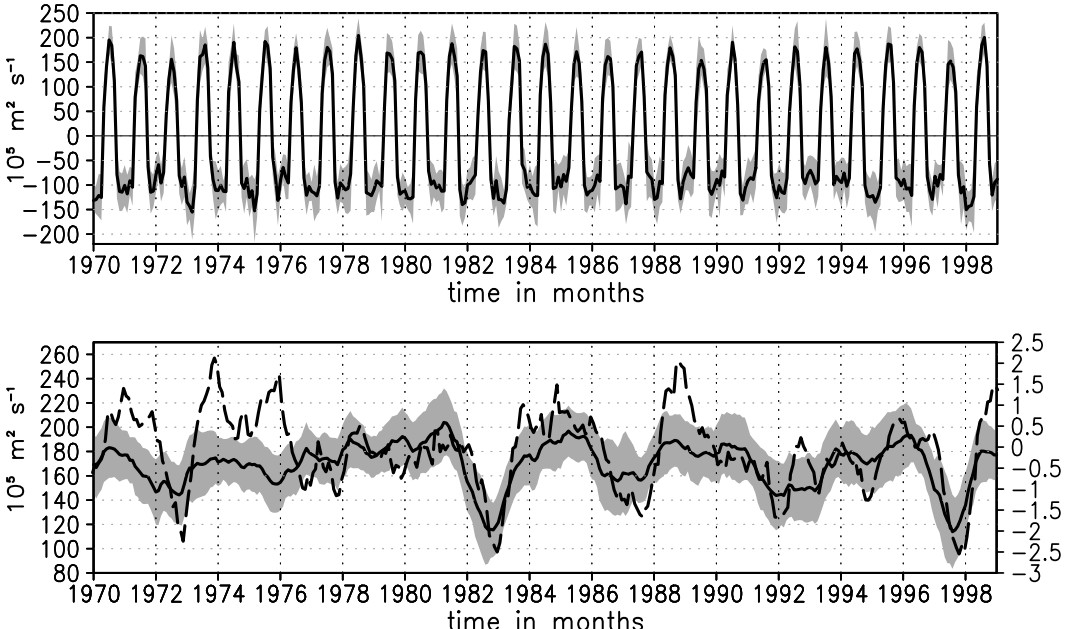

**Figure 1.** Time series of indices for the period 1970–1999. Top: index for the monsoon circulation (MIDX); bottom: index for the Walker circulation (solid) and for comparison the Niño3.4 index derived from the HadISST1 dataset (dashed), multiplied by −1 for better comparison. Grey shadings indicate one standard deviation of the multi–model mean statistics.

## 3   Methods

### 3.1   Climatology of the AMA

We derive climatologies to assess the average impact of the ASM on the upper tropospheric circulation, temperatures, $H_2O$ and $O_3$ mixing ratios, by means of multi–model averages (MMOD). By nature, the MMOD will graduate the occasionally

5   large differences among individual CCMs, in comparison to the re-analysis and satellite observation. Characteristic quantities derived as box averages, or extreme values within a certain area related to the monsoon anticyclone are derived for the individual models to assess the model spread of the CCMs, in comparison to the MMOD and the observational reference. The spread in statistics for individual CCMs is an indication for the robustness of the MMOD.

### 3.2   Interannual variability of the AMA

10   The variability of the temperatures and the $H_2O$ and $O_3$ mixing ratios is assessed with a multiple linear regression model, to estimate the relative importance of the ASM circulation, ENSO and the QBO for these quantities in the UTLS.


### 3.2.1 Separating tropical circulations

To quantify the inter–annual variability of the ASM circulation, a monsoon circulation index is calculated as described in Tanaka et al. (2004). The method is based on the separation of the horizontal flow in the upper troposphere. According to the Helmholtz theorem, the horizontal flow can be separated into a rotational, nondivergent ($\mathbf{v_{h/r}}$) and a divergent, irrotational

component ($\mathbf{v_{h/d}}$): $\mathbf{v_h} = \mathbf{v_{h/r}} + \mathbf{v_{h/d}}$, with $\nabla \cdot \mathbf{v_{h/r}} = 0$ and $\nabla \times \mathbf{v_{h/d}} = 0$. This allows to express the horizontal flow by a combination of streamfunction $\psi$ and velocity potential $\chi$ in the following way: $\mathbf{v_h} = \mathbf{k} \times \nabla \psi + \nabla \chi$.

Following Tanaka et al. (2004) tropical circulations (Hadley-, Walker-, and monsoon circulation) can be identified by further separating the velocity potential $\chi$.

$$\chi(t,x,y) = [\chi(t,y)] + \bar{\chi}^*(x,y) + \chi^{*\prime}(t,x,y), \tag{1}$$

where in a first step $\chi(t,x,y)$ is separated into the the zonal mean $[\chi(t,y)]$, representing the Hadley circulation, and the eddy component $\chi^*(t,x,y)$. The eddy component can be further separated into a time mean component $\bar{\chi}^*(x,y)$, representing the Walker circulation and a transient component $\chi^{*\prime}(t,x,y)$, representing the monsoon circulation.

During the ASM season, the strength of $\chi^{*\prime}(t,x,y)$ can directly be related to the intensity of the ASM, with strong upper tropospheric divergent flow where the most intense convective systems are located. As focus is on the influence of the ASM on

the UTLS, we chose the velocity potential at 150 hPa to derive an index for the monsoon circulation (MIDX) close to the lower stratosphere. The MIDX is defined as the maximum in monthly mean $\chi^{*\prime}(t,x,y)$ located over south–east Asia from May to September, and the minimum over the same area during the remainder of each year. Due to this definition the MIDX changes from positive values during May–September to negative values during October–April, when the upper tropospheric flow over south–east Asia is convergent (Fig. 1). A time varying Walker circulation index (WIDX) is produced as the time mean of the

eddy component of the velocity potential $\bar{\chi}^*(x,y)$, using the running mean of twelve individual months. The values of the WIDX are defined as the maximum in monthly mean $\bar{\chi}^*(x,y)$ over the western Pacific. During ENSO warm events the Walker circulation weakens, which is reflected in a decreasing WIDX. A comparison of the WIDX (multi–model average) with the Niño3.4 index derived from the HadISST1 dataset (Fig. 1) shows some similarities, especially for the strongest ENSO warm events during 1982/83 and 1997/98, whereas the Niño3.4 index describes larger ENSO variability. As all CCMs prescribe SSTs

of the HadISST1 dataset, they react in a similar way with respect to the Walker circulation.

### 3.2.2 Multiple linear regression model

To identify the temperature and trace gas changes of the lowermost stratosphere associated with the ASM circulation, ENSO, and the phase of the QBO we use a multiple linear regression (MLR) model, as described in SPARC CCMVal (2010). The following basis functions are applied: a constant offset, a linear trend, the QBO, the MIDX, the Niño3.4 index, the 10.7cm

solar flux, and basis functions for three major volcanic eruptions (Agung, El Chichón, and Pinatubo), that are realized by using an idealized function with a rapid increase and an exponential decay (Bodeker et al., 1998). The time series of the MIDX are calculated separately for each CCM and the ERA-Interim data according to Tanaka et al. (2004), as described in Section 3.2.1.





The QBO basis function consists of the time series of the zonal mean zonal wind in 50 hPa averaged over the two innermost tropical latitudes, derived for each individual CCM simulation and the ERA-Interim data. The regression model contains in addition an orthogonal version of the 50 hPa QBO, to account for the fact that within the vertical range of the QBO two distinct phases are present. The Niño3.4 index is calculated as an area averaged, standardised anomaly of the HadISST1 SST for the Niño3.4 region 170–120°W, 5°S–5°N. An alternative ENSO index is derived with the WIDX (see Sec. 3.2.1). Because the WIDX is strongly correlated with the Niño3.4 index for the ERA-Interim data ($r = 0.7$), only the Niño3.4 index is included in the MLR. The regression model is applied to time series of $n$ JA averages. The trend and the long–term average are removed for the basis functions $QBO(t)$, $QBO\_orthog(t)$, $MIDX(t)$, $N34(t)$, and $solar(t)$.

$$
\begin{aligned}
y(t) =\ & \beta_{offs} offset + \beta_{tr} trend(t) + \\
& \beta_{qbo} QBO(t) + \beta_{qbo\_or} QBO\_orthog(t) + \\
& \beta_{midx} MIDX(t) + \beta_{n34} N34(t) + \beta_{sfl} solar(t) + \\
& \beta_{vol} Volcano(t) + \varepsilon(t), \quad t = 1, n
\end{aligned}
\tag{2}
$$

The regression equation 2 models the time series of a quantity $y(t)$ by linearly fitting the time series of the basis functions to it by means of least squares estimates, resulting in the fit parameters $\beta_j$, and a residual $\varepsilon(t)$. Results of the MLR are discussed in Section 5, for the fit parameters $\beta_{midx}$, $\beta_{n34}$, and $\beta_{qbo}$, that are multiplied by a factor of 1.0, 2.5, and 25.0 respectively to account for a typical amplitude of the proxy time series. Please see appendix A for the calculation of significances, and appendix B for information about the treatment of the autocorrelations.

## 4   The climatological state of the Asian Summer Monsoon during July and August

The diabatic heating associated with the convective systems of the ASM causes a divergent outflow in the UTLS and eventually, as a large–scale organised response, the anticyclone (e.g. Garny and Randel, 2013). Liu et al. (2007) highlighted the role of diabatic heating over the Tibetan Plateau in generating a minimum in potential vorticity, coincident with the AMA. The AMA has been recognised to influence the transport pathways of various trace gases (e.g. CO, $CH_4$, $H_2O$, HCN), entering the stratosphere in the tropical UTLS (Fu et al., 2006; Randel et al., 2010; Wright et al., 2011). As pointed out by Goswami et al. (1999), the northward migration of strong convective activity during the ASM leads to a regional reversal of the Hadley circulation, with ascent near 20°N and descent near the Equator. This is partly, on the eastern flank of the AMA, a manifestation of the strong anticyclonic circulation. In this section the climatological features of the ASM in the CCMs are assessed by comparing the multi–model mean (MMOD) of their JA average circulation and $H_2O$ and $O_3$ mixing ratios in the UTLS with ERA-Interim and MIPAS data.



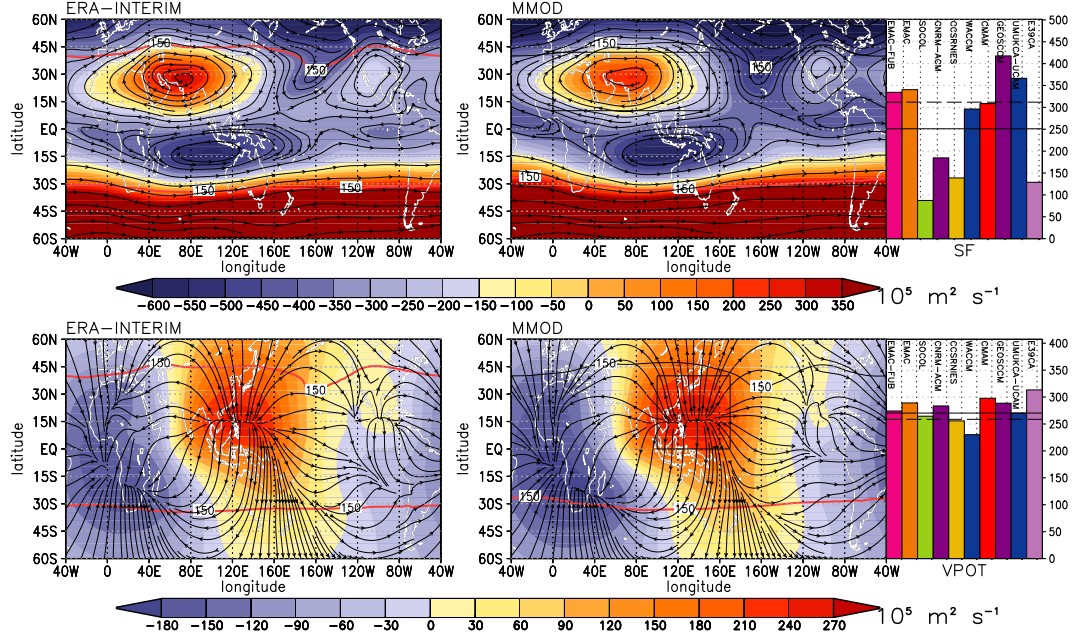

**Figure 2.** Top: long-term monthly mean stream function (in $10^5$ m$^2$ s$^{-1}$) for JA at 150 hPa in $60°$S – $60°$N; left: ERA-Interim (35 years); right: the multi-model average (45 years). Streamlines of the rotational horizontal wind. Indicated with the red solid contour is the intersect of the tropopause with 150 hPa. The maximum of individual models within the area marked by the black rectangle is displayed as bar chart, where the solid horizontal line represents the multi-model average and the dashed horizontal line represent the maximum of ERA-Interim. Bottom: as above but for the velocity potential (in $10^5$ m$^2$ s$^{-1}$) with streamlines of the divergent horizontal wind.

## 4.1 The monsoon anticyclone and related zonal and meridional flow

The divergence-free part of the horizontal flow is described with the stream function (Sec. 3.2.1). Figure 2 (top) shows that during the mature phase of the ASM the horizontal flow in the UTLS over southern Asia is dominated by an anticyclonic stream function, extending from $40°$W – $160°$E in the longitudinal and from the Equator – $50°$N in the latitudinal direction.

5  A second anticyclone exists over North America, related to the North American monsoon. The large values of the cyclonic stream function in the southern hemisphere (more than $350 \times 10^5 m^2 s^{-1}$) are associated with the polar vortex. The bar chart on the right of Figure 2 (top) shows the maximum climatological JA stream function for each model within the region indicated by the black rectangle in Figure 2 (top). The CCMs show a large spread in their maximum stream function values. Four CCMs strongly underestimate the ERA-Interim stream function, leading to a weaker MMOD stream function than in ERA-Interim. As

10  explained, the divergent part of the upper tropospheric circulation can be described by the velocity potential $\chi$ (Fig. 2, bottom). During JA $\chi$ has the largest positive values centred near $15°$N in the western Pacific. The stream lines in Fig. 2 (bottom) show the divergent horizontal flow, directed from the maximum $\chi$ towards the minimum, extending from the gulf of Guinea to southern Africa. The positive peak values in $\chi$ are thought to be related with regions of upwelling, coinciding with the onset of



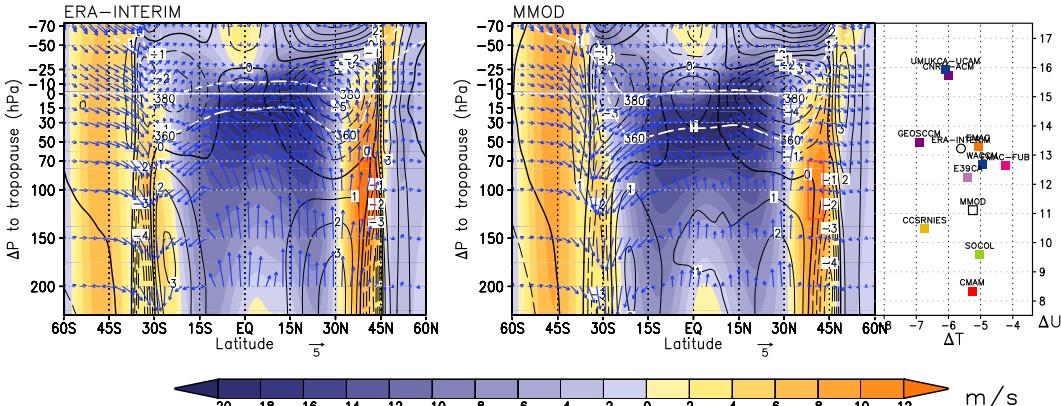

**Figure 3.** Latitude–height sections of JA long-term mean anomaly of the zonal velocity averaged over 30° in longitudes, centred where the 150 hPa eastward directed divergence free zonal wind maximizes, from the zonal average (shaded). All data interpolated to pressure levels relative to the tropopause height. Left: ERA-Interim (35 years); right: the multi-model average (45 years). Black contours show the temperature anomalies of the respective latitude sections from the zonal average. White dashed contours indicate the 360 and 380 K isentropic levels. Blue arrows denote the meridional (in m/s) and vertical velocity (in mm/s). The maximum of the zonal wind anomaly and the minimum of the temperature anomaly near the tropopause are displayed as scatter plot for individual models (squares), the multi-model average, and ERA-Interim (circle).

the divergent winds, and vice versa for the negative peak values in $\chi$. There is quite good agreement between the ERA-Interim data and the MMOD in the location and strength of the up– and downwelling, and also the maxima of the individual CCMs in the region of upwelling, indicated by the bar chart on the left side of Fig. 2 (bottom), have only relative small deviations from the MMOD.

The regional anticyclonic circulation is a large deviation from the zonal mean. To distinguish between tropospheric and stratospheric levels and their interactions in the AMA region, the analysis of the cross sections (Fig. 3 and 4) is done with data interpolated to pressure levels relative to the tropopause pressure. Shown are deviations from the zonal average of the zonal wind component (shaded), and the temperature (contoured). To account for differences among the CCMs in the location of the AMA, we show the deviations of an average that spans 30° in longitude, centred at the longitude where the 150 hPa eastward

directed divergence-free zonal wind maximizes. There is a strengthening of the zonal wind component on the northern flank of the AMA with a maximum near 40°N (Fig. 3). Regional averages to the west and the east (Fig. 4) of the AMA, show anomalies in the meridional wind component, which are a consequence of the anticyclonic circulation. As for Figure 3 the regions span 30° in longitude, but for Figure 4 are centred where the northward (southward) directed divergence free meridional wind maximizes at the western (eastern) edge of the AMA. Centred at the tropopause and 30°N, negative temperature anomalies in

the UTLS are present in all three selected regions, i.e. at the centre of the AMA (Fig. 3) and the western and eastern edges (Fig. 4). This lower than average cooling within the AMA is related to the convective activity during the ASM (Park et al., 2007), but the negative temperature anomaly located in the core region and the eastern flank of the AMA can also be due to





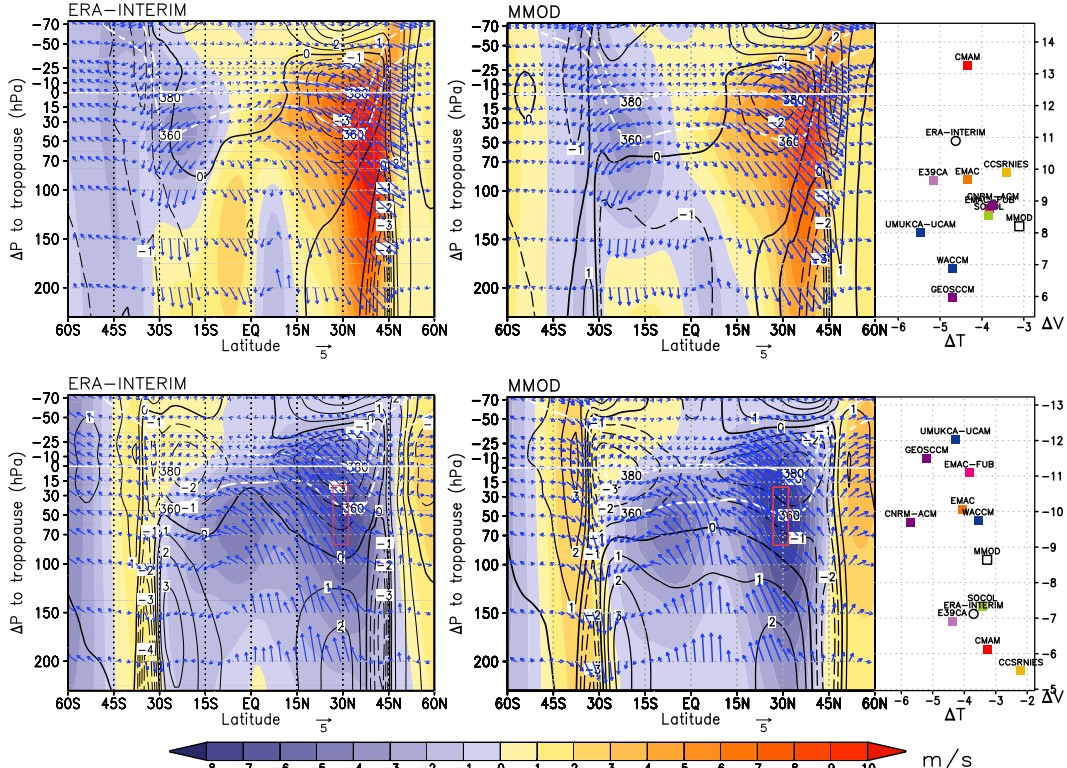

**Figure 4.** As Fig. 3 but for the meridional velocity averaged over $30°$ in longitudes; top: centred at the western flank of the AMA where the 150 hPa northward directed divergence free meridional wind maximizes, relative to the zonal average; bottom: centred at the eastern flank of the AMA where the 150 hPa southward directed divergence free meridional wind maximizes, relative to the zonal average.

adiabatic cooling in regions of uplift (s. Fig. 3 and 4 bottom) above the convective systems. At the western flank of the AMA, negative temperature anomalies are prevailing as well, despite the downward directed vertical velocity, that implies adiabatic warming. The 360 K isentropic level, located in the UT at lower latitudes and in the LS at higher latitudes, intersects the tropopause near $42°$N. The 380 K isentropic level is located in the stratosphere except for a small region near $35°$N, where it is

5  intersecting the tropopause. The behaviour of the isentropes, reaching higher pressures within the monsoon region, is reflecting the larger energy content of the ASM region. To visualise the spread of the CCMs relative to the MMOD and the ERA-Interim, the temperature anomalies and the anomalies of the zonal (Fig. 3) and meridional (Fig. 4) wind components are displayed as scatter–plot on the right side of each figure. The circulation and temperature anomalies simulated by the individual CCMs all point to the same direction, although there exists considerable spread with regard to the strength of the anomalies.

10  **4.2  Mean temperatures, water vapour and ozone mixing ratios in the UTLS**

$H_2O$ mixing ratios on isentropic levels show large values confined to the center of the AMA at 360 K, which spread out to the north–west and east of the AMA at 370 and 380 K (Fig. 5). The low values of the observed outgoing longwave radiation ($< 205$





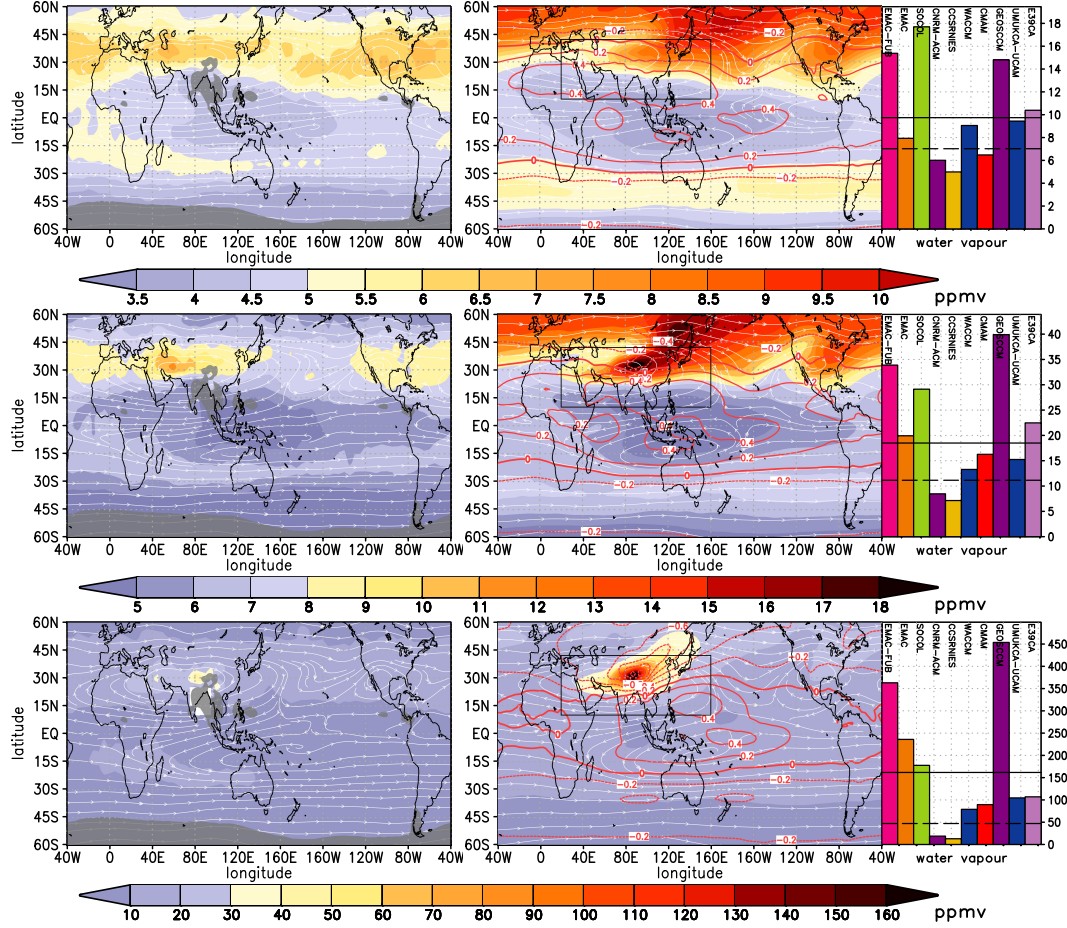

**Figure 5.** Long-term monthly mean $H_2O$ mixing ratios in ppmv for JA at 360, 370, and 380 K (from bottom to top) for latitudes from $60°S – 60°N$; left: MIPAS (2003, 2005–2011), overlaid grey shaded regions where the OLR $\leq 205$ W m$^{-2}$ ; right: the multi-model average (45 years). Overlaid as streamlines are the horizontal wind components; red contours indicate the net radiative heating rates in K day$^{-1}$. The maximum of individual models within the area marked by the black rectangle is displayed as bar chart, where the solid horizontal line represents the MMOD and the dashed horizontal line represents the maximum of MIPAS.

W m$^{-2}$), as indicated by the grey shading, identify the BoB and the western coast of Myanmar to be the region of the strongest convection during JA. The MIPAS $H_2O$ maximum at 360 K is located north–west of the region with strongest convection, and at higher levels (370 and 380 K) the maximum is even farther away from its supposed source region. In the CCMs $H_2O$ mixing ratios show a similar behaviour, although the values are much higher and tend to spread out more to the north–east.

5  To indicate the potential for further vertical uplift, the net diabatic heating rate is shown on isentropic levels. Positive values indicate ascent to levels of higher potential temperature. The mean radiative heating rates of the CCMs at the 360 K level are only positive at latitudes south of $\sim20°N$ (including the BoB). This indicates the possibility of slow uplift with upward transport of tracers in this region. For the region of the $H_2O$ maximum, centred around $30°N$ at 360 and 370 K, however, the




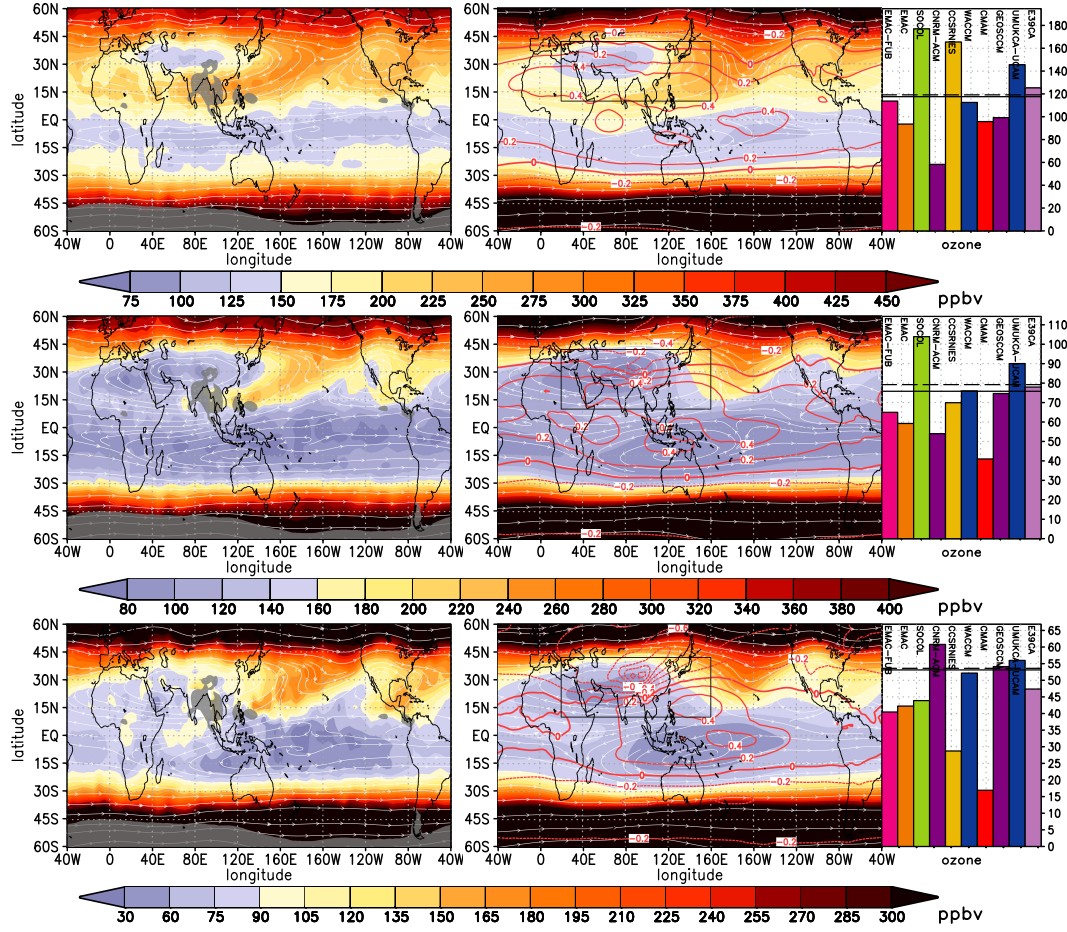

**Figure 6.** As Fig. 5 but for ozone in ppbv. The minimum of individual models within the area marked by the black rectangle is displayed as bar chart, where the solid horizontal line represents the MMOD and the dashed horizontal line represents the minimum of MIPAS.

net radiative heating rates are negative, indicating descent. This highlights the important role of horizontal transport within the AMA, moving tracers away from the regions of convective outflow.

The signatures of the monsoon circulation in UTLS tracer extrema, derived here from monthly mean CCM and MIPAS data as one single extremum, exhibit a considerable day-to-day variability related to the variability of the AMA. The anticyclone

5  often splits, with one centre over Iran and a second centre over China (Garny and Randel, 2013). The AMA is thus a quite dynamic system, with intra-seasonal variability also in the tracer concentrations of the UTLS.

At 360 and 370 K the MIPAS $H_2O$ mixing ratios are still isolated within the AMA, in contrast to the CCMs, and the maxima in $H_2O$ mixing ratios in the AMA decrease from 48 (360 K) to 11 ppmv (370 K) with increasing height (dashed line in the bar chart of Fig. 5). The lower level (360 K) is influenced by the latent heat release from convective activity above the TP, as

10  indicated by the temperature maximum north of the centre of the AMA, which is present in both, ERA-Interim and the multi–





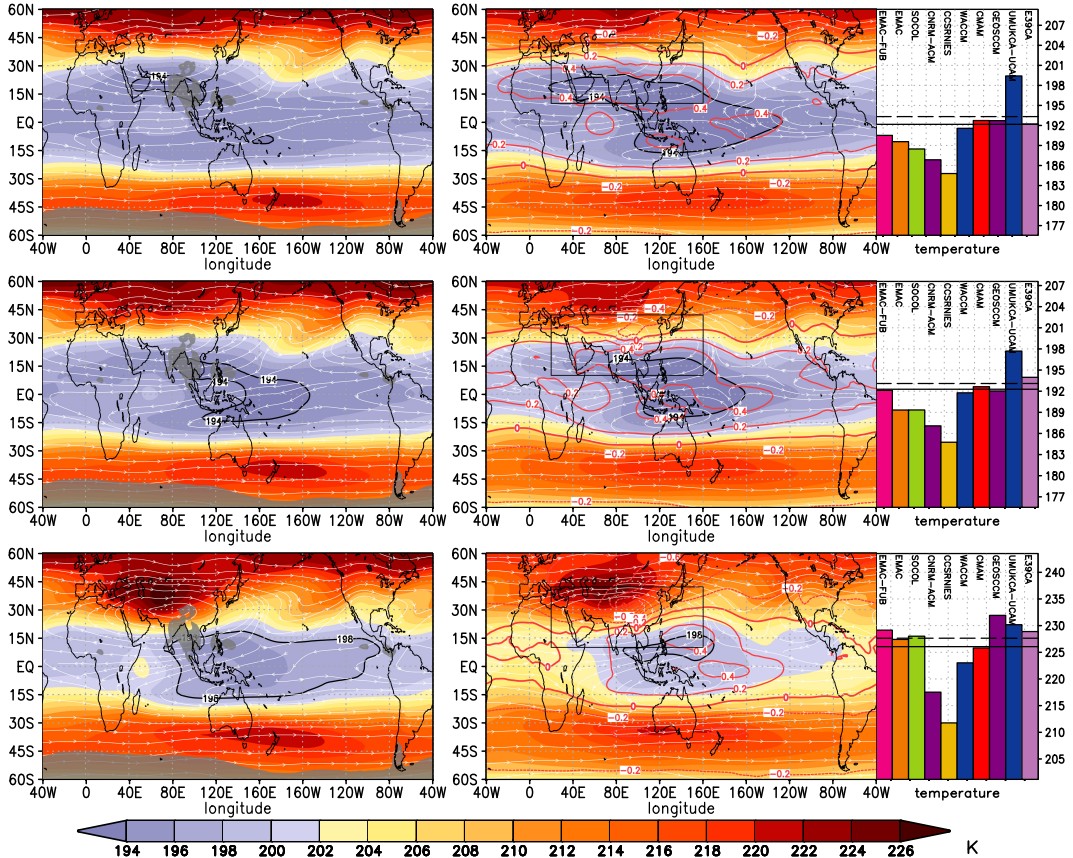

**Figure 7.** As Fig. 6 but for temperature. Left column: JA ERA-Interim temperature averaged from 1979–2014. Contour lines for the lowest temperatures of the multi-model average are given in black: 296 K (at 360 K), 294 K (at 370 and 380 K). At 370 and 380 K the minimum (at 360 K the maximum) of individual models within the area marked by the black rectangle is displayed as bar chart, where the solid horizontal line represents the MMOD and the dashed horizontal line represents the value for ERA-Interim MIPAS.

model average (Fig. 7). With increasing height the local temperature minimum is developing within the AMA and extending northward. This temperature structure in the ASM region is also obvious in the latitude–height sections (s. Fig. 3 and 8).

Between 360 and 380 K dehydration reduces the $H_2O$ mixing ratios to 7 ppmv but this maximum is not as well pronounced as at lower levels. Most CCMs overestimate the $H_2O$ mixing ratios, some by more than a factor of three at 370 K. With only two

5  CCMs underestimating the $H_2O$ mixing ratios within the AMA, the MMOD of the maximum mixing ratio reaches 18 ppmv (solid line in the bar chart of Fig. 5) and more than 9 ppmv at 380 K. $H_2O$ transport occurs to higher latitudes, as can be seen from the MMOD of the CCMs, but this is not confirmed by the MIPAS data which show lower $H_2O$ mixing ratios northward of 45°N. The MMOD of the CCMs shows strong coherence between the northern hemispheric temperature structures at 370 and 380 K (Fig. 7) and the corresponding $H_2O$ mixing ratio fields, with the exception of the ASM and the North American





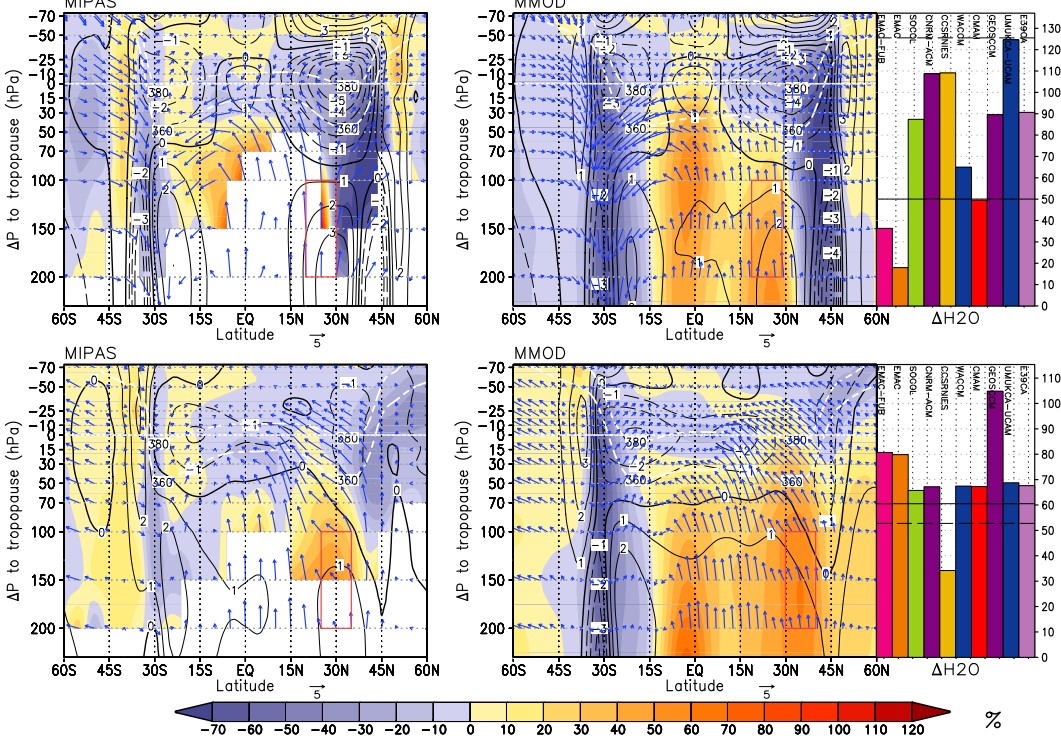

**Figure 8.** Latitude–height sections for pressure levels relative to the tropopause pressure, from the $60°S – 60°N$ for the multi-model JA long-term mean (45 years) anomaly of water vapour mixing ratios in % averaged as in Fig. 3 (top), and averaged between $120° – 160°E$ (bottom) relative to the zonal average, shading interval is 10%. Black contours show the temperature anomalies of the latitude sections from the zonal average. White dashed contours indicate the 360 and 380 K isentropic levels. Blue arrows denote the meridional (in m/s) and vertical velocity (in mm/s). The maximum of the $H_2O$ anomaly within the region enclosed by the red rectangle for individual models is displayed as bar chart, where the solid horizontal line represents the multi-model average, and the dashed horizontal line represents the maximum of the MIPAS data.

monsoon regions that serve as a source region for $H_2O$. As indicated by the bar chart of Figure 5, the individual CCMs show large deviations in $H_2O$ extrema from the MMOD in the ASM region, largest at 360 K and less pronounced at higher levels.

A particular feature of the UTLS above the ASM are the low $O_3$ mixing ratios confined within the AMA, which are caused by upwelling of lower tropospheric air (Randel and Park, 2006; Park et al., 2007). As shown by Braesicke et al. (2011) with GCM simulations using prescribed lower or higher $O_3$ mixing ratios within the AMA, lower $O_3$ mixing ratios have the tendency to strengthen, and cool the AMA and vice versa. The MIPAS data show low $O_3$ mixing ratios at all three isentropic levels (Fig. 6) with the most pronounced ozone minimum at 370 K. Similar to the $H_2O$ maximum, the location of the MIPAS $O_3$ minimum is northwest of the region of the most intense convective activity. Unlike for $H_2O$, there is quite good agreement of the MMOD $O_3$ mixing ratios with the MIPAS data. $O_3$ in the UTLS can better serve as a passive tracer than $H_2O$, due to its relatively long lifetime in the UTLS, and as it is not affected by dehydration. As indicated by MIPAS $O_3$ data, tongues of air with high $O_3$





mixing ratios are transported on the eastern flank of the AMA towards lower latitudes and form a ring of high $O_3$ mixing ratios around the centre of the AMA at 360 K.

In Figure 8 mean anomalies in two regions of the AMA are shown for $H_2O$ mixing ratios, temperature, meridional and vertical transport. The first sectional average, as for Figure 3 represents the core region of the AMA, whereas the second region
is averaged over fixed longitudes from $120°$–$160°$E, representing the eastern edge of the AMA where meridional winds are southward and uplift prevails. Two regions with enhanced $H_2O$ mixing ratios are present in both sectional averages, one centred near the Equator and the other near $30°$N. The MMOD of the eastern sectional average shows $H_2O$ enhancement, reaching the equatorial stratosphere, coincident with an upward and southward directed meridional circulation. Although the general features of the CCMs, with two regions of enhanced $H_2O$ mixing ratios, is also visible in the MIPAS data, the direct link of
$H_2O$ transport into the stratosphere, as suggested by the MMOD, cannot be confirmed by the MIPAS data.

### 4.3   Upper tropospheric monsoon circulation

As described in Section 3.2.1 the horizontal flow can be separated into a a rotational part and a divergent part, the velocity potential. It can be further separated into a time mean component and a transient component, representing the Walker and monsoon circulations respectively (Tanaka et al., 2004)).
Figure 9 shows the separation of the velocity potential into a component representing the Walker circulation (Fig. 9, top) and a component representing the monsoon circulation (Fig. 9, bottom). The position of the maximum upwelling for individual years is marked with open circles, where the size of the circle represents the strength of the upwelling. For the annual average the region of strongest upwelling, related to the Walker circulation (Fig. 9, top), is located over the equatorial western Pacific, in most years to the north–west of New Guinea. The main region of downwelling is extending from the central Sahara to western
Africa, whereas a secondary centre of downwelling is located off the coast of Peru.

The centre of the strongest upwelling related to the monsoon circulation (Fig. 9, bottom) is located over south–east Asia, with the centres of individual years located in a region extending from the BoB to eastern China. This area partly overlaps with the region of the lowest OLR (see Fig. 5) and therefore the monsoon circulation index, derived from the maxima in the seasonal decomposition of the velocity potential, is a good indicator for the overall strength of the monsoon circulation. Although
the ERA-Interim climatological average of both decompositions, annual average associated with the Walker circulation and seasonal average associated with the monsoon circulation, is slightly lower than for the MMOD, the shape and the locations of the maxima for individual years in the CCMs is quite similar.

## 5   Interannual variability of temperature, $H_2O$ and $O_3$ in the UTLS

So far we have characterised the climatological behaviour. Here, we characterise the dominant internal modes of inter annual
variability like ENSO, the QBO, or the monsoonal variability. Although there is evidence of a coupling between the ASM and ENSO through the Walker circulation (e.g., Webster and Yang, 1992; Ju and Slingo, 1995), we make an attempt to separate the influence of the ASM and ENSO on the UTLS temperatures, the $H_2O$, and $O_3$ mixing ratios by applying a multiple linear





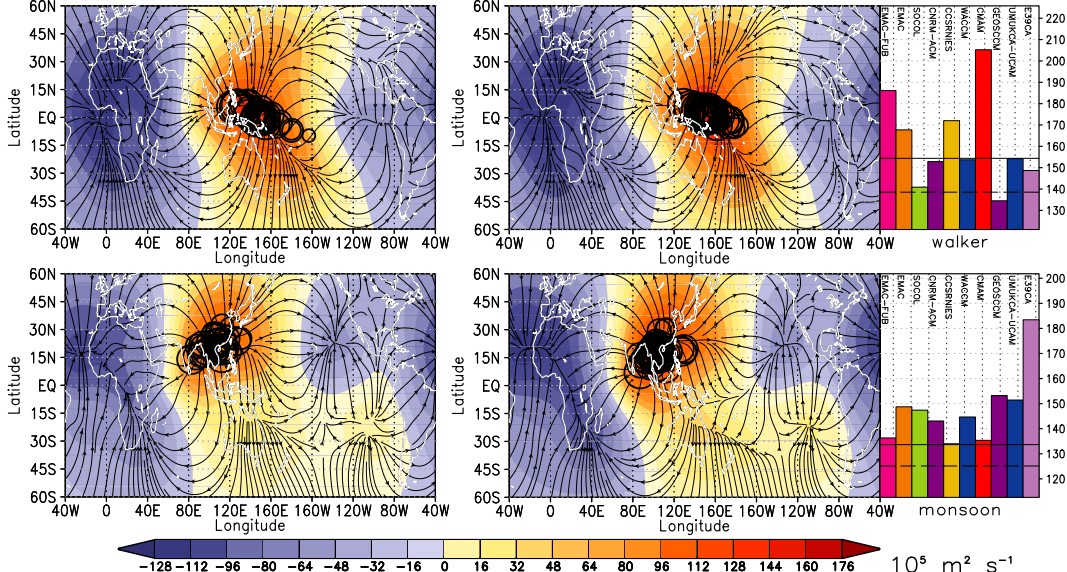

**Figure 9.** Top: Annual mean climatology of the MMOD velocity potential at 150 hPa after subtracting the zonal average, representing the divergent flow associated with the Walker circulation; (left) for ERA-Interim (1979–2014); (right) for the multi–model average of the CCMs. Bottom: JA climatology of the velocity potential at 150 hPa after subtracting the zonal average and annual average, representing the divergent flow associated with the ASM. Black circles mark the position of the maximum velocity potential for individual years. The bar charts on the right indicate the maximum climatological velocity potential for the individual CCMs; the solid horizontal line represents the MMOD and the dashed horizontal line represents the maximum of ERA-Interim.

regression model as described in Section 3.2.2. We analyse the influence of the monsoon circulation, ENSO and QBO on the transport characteristic of the AMA.

The regression coefficients of the individual CCMs are combined by a simple average for the CCMs listed in Table 1. The results of the $t$-tests of the multiple linear regression results for the individual CCMs are combined by using the Z–transform
5   method (Stouffer et al., 1949; Whitlock, 2005). Regions where the combined regression coefficients are not significant are marked by grey shading, overlaid on the colour shading used to emphasize the regions with the largest regression coefficients (s. Appendix A for more details).

### 5.1 Influence of the monsoon circulation

The influence of the strength of the Asian summer monsoon on $H_2O$, and $O_3$ mixing ratios in CCMs and ERA-40 data has been
10   analysed in Kunze et al. (2010) by separating the data according to the monsoon Hadley index (MHI) in weak and strong ASM seasons. During stronger ASM seasons, $H_2O$ and $O_3$ were found to be anticorrelated, with lower $O_3$ and higher $H_2O$ mixing ratios within the AMA, as a result of stronger convective activity during stronger ASM seasons. In contrast to the MHI, the now used MIDX is a more direct measure for the ASM strength. Randel et al. (2015) used a different approach in distinguishing





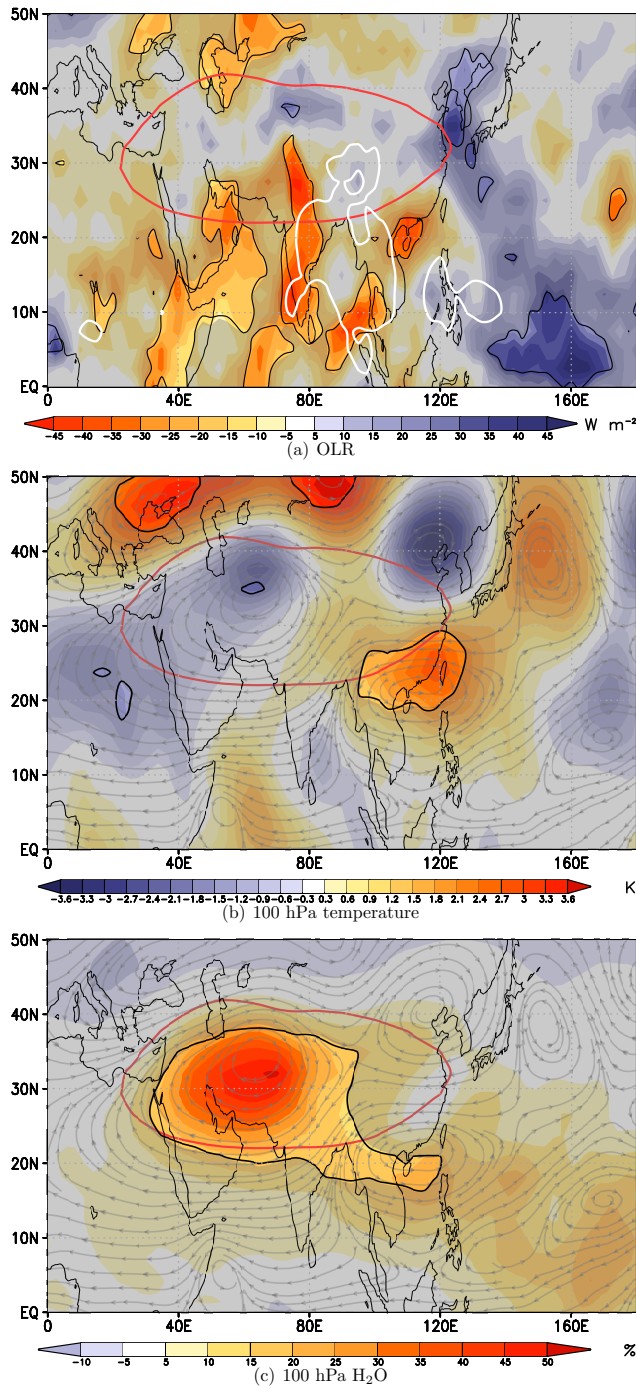

**Figure 10.** Regression coefficients from MLR for the MIDX of JA average data, with (a) NOAA OLR, (b) ERA-Interim 100 hPa temperature, (c) ERA-Interim 100 hPa $H_2O$. The 205 W m$^{-2}$ contour line of NOAA OLR is plotted in white on (a). Red contours of the 16750 m geopotential height at 100 hPa mark the position of the AMA. Grey streamlines on (b) and (c) mark the horizontal wind components, regressed on the MIDX. Overlaid grey shading indicates regions where the regression is not significant at the 95% level.



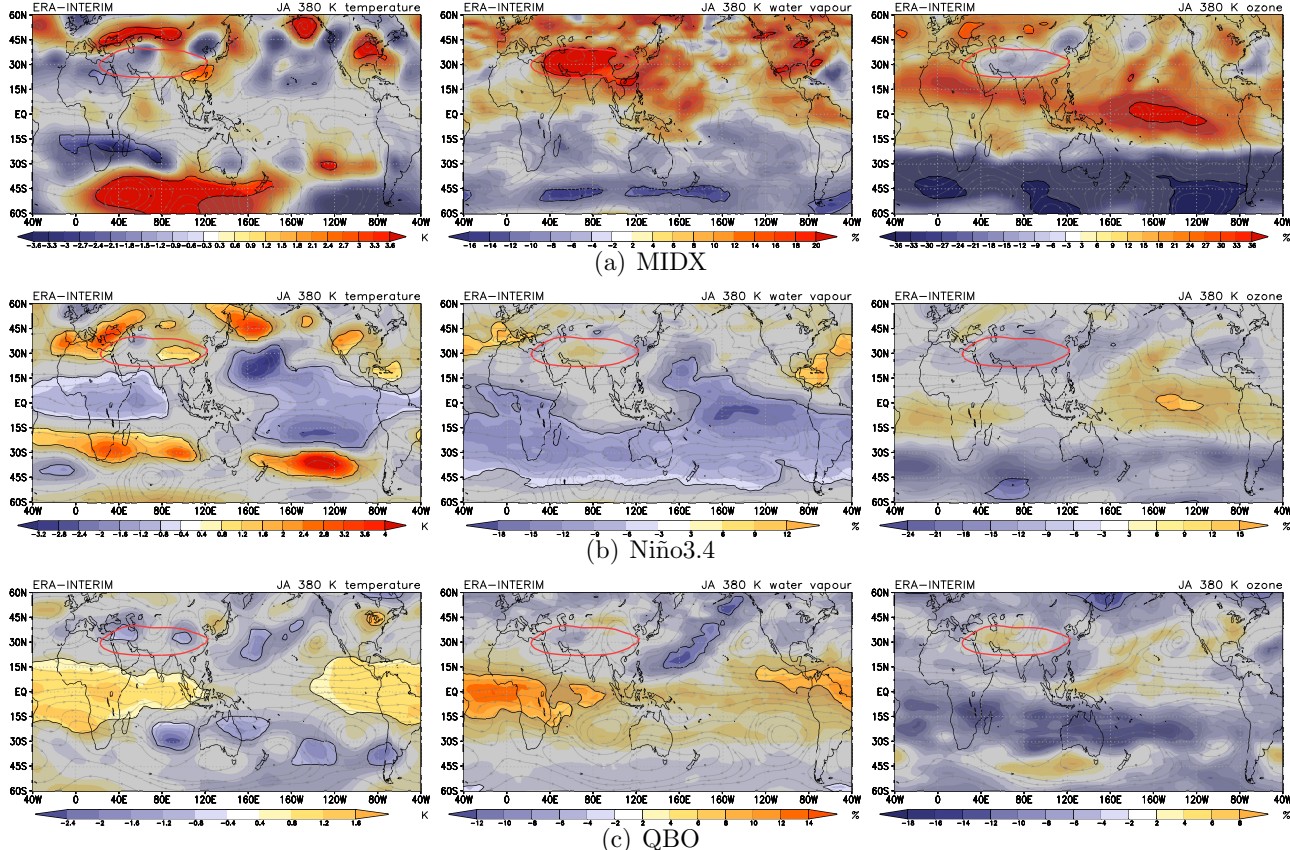

**Figure 11.** Regression coefficients from MLR of ERA-Interim data, for (a) the monsoon circulation index, (b) the Niño3.4 index, (c) the QBO time series, with temperature (left), water vapor (middle) and ozone (right) on the 380 K isentropic level. The regression coefficient for the $H_2O$ and $O_3$ mixing ratios are displayed in % of the long–term average of the respective JA average. The red contour level for the 380 K Montgomery streamfunction of $3625 \times 10^2$ m$^2$ s$^{-2}$ marks the position of the AMA. Overlaid grey shading marks regions where the regression is not significant at the 95% level.





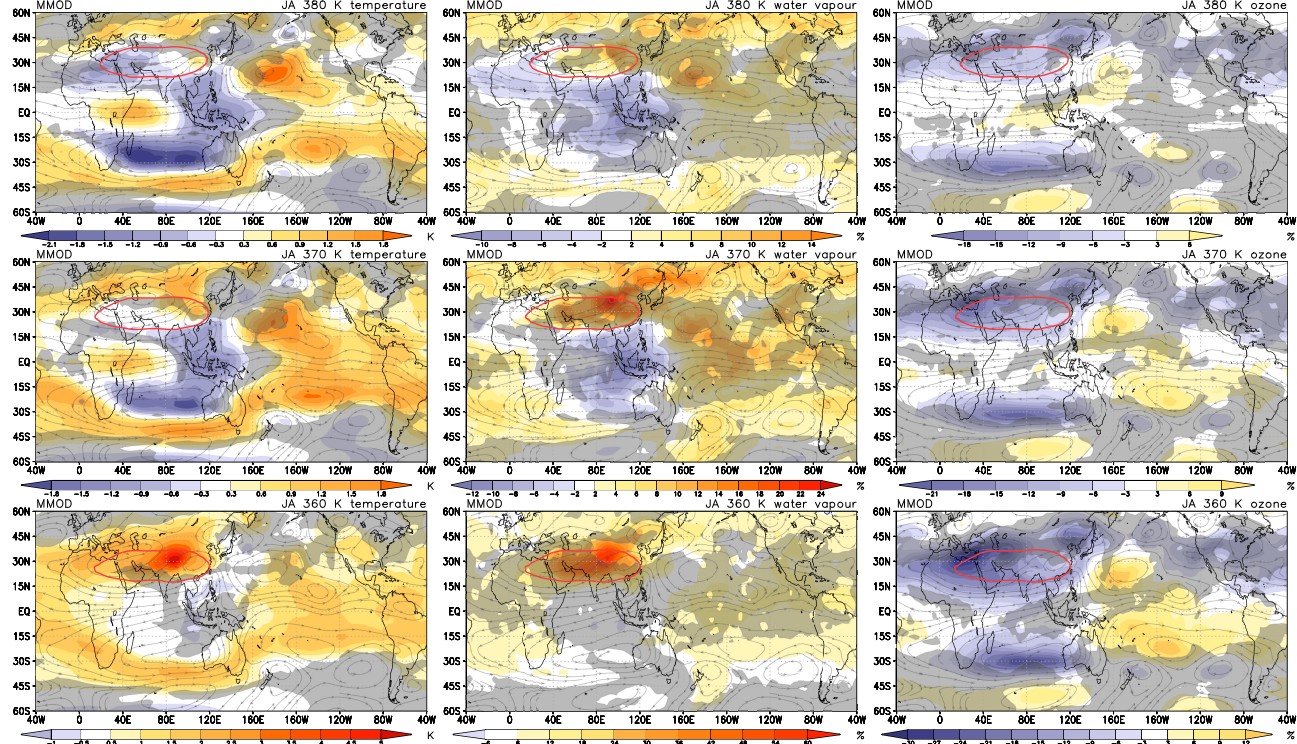

**Figure 12.** Multi model average of the CCMs of the regression coefficient of the monsoon circulation index (MIDX) with temperature (left), $H_2O$ (middle) and $O_3$ (right) on isentropic levels 360, 370, and 380 K (from bottom to top). The regression coefficients of the $H_2O$ and $O_3$ mixing ratios are displayed in % of the long–term average of the respective JA average. Grey streamlines show the horizontal wind components, regressed on the MIDX. The red contours of the Montgomery streamfunction (360 K: 3513, 370 K: 3570, 380 K: 3625, in $10^2$ $m^2\ s^{-2}$) mark the position of the AMA. Overlaid grey shading marks regions where the regression is not significant at the 97.5% level.

between wet and dry phases in the AMA, to identify the conditions leading to either dry or wet $H_2O$ extrema. During the wet phase they found reduced convection over the regions of strongest convective activity. The corresponding temperature anomalies, linked to reduced convection, show a dipole structure with warmer conditions on the southern edge of the AMA during wet phases and vice versa. They therefore conclude that during weaker ASM seasons less effective dehydration in the warm anomaly at the southern edge of the AMA is responsible for the higher $H_2O$ mixing ratios within the AMA.

In order to further elucidate the role of the monsoon intensity on UTLS temperature, $H_2O$ and $O_3$ content, we perform a MLR analysis using the MIDX. As MIDX is a direct measure of the strength in upwelling, the regression patterns represent the changes due to an increase in ASM activity. Positive regression coefficients in strong monsoon years (MIDX $\gg 0$) indicate regions that tend to be warmer or have increased $H_2O$ or $O_3$ mixing ratios. Negative regression coefficients indicate regions where strong monsoons lead to cooling or decreased $H_2O$ or $O_3$ mixing ratios.



The MIDX regression coefficients for the JA average NOAA OLR (Fig. 10a) show a similar pattern as shown by Randel et al. (2015) (their Fig. 5a) for the wet case with a significant decrease in OLR, indicating colder cloud tops, i.e. stronger convective activity, over the Indian subcontinent and the Arabian sea. For stronger monsoon seasons we also get an increase in convective activity over the BoB, the western coast of Myanmar, and Taiwan, which is in contrast to the results of Randel et al. (2015).

This decrease of OLR with monsoon activity is partly coincident with the regions of strongest convection, indicated by the $OLR < 205$ W m$^{-2}$ (white contour line in Fig. 10a). The MIDX regressed on ERA-Interim 100 hPa temperatures reproduces the anomalous dipole temperature structure with lower temperatures at the northern and higher temperatures at the southern edge of the AMA, but the patterns are more similar to the inverse of that shown by Randel et al. (2015) (their Figure 8b) for the dry case, with a significant warming to the southeast of the AMA, and two cold anomalies located on the western side and on

the northeastern flank of the AMA (Fig. 10b). From our analyses for the MIDX regression on ERA-Interim temperatures and $H_2O$ mixing ratios (Fig. 10c) we conclude that during a stronger monsoon season $H_2O$ mixing ratios should increase within the AMA. The reason for the discrepancy with Randel et al. (2015) may be related to the different approach. Their analyses work the other way round, based on the wet and dry anomalies confined in the AMA, and the associated temperature and OLR anomalies, whereas we are obtaining the wet anomaly as a result of the regression with the MIDX.

The analyses on the 380 K isentropic surface (Fig. 11a) show similar anomaly patterns within the Asian monsoon region as analysed for 100 hPa. The MIDX regressed on the ERA-Interim $H_2O$ and $O_3$ mixing ratios shows a significant increase in $H_2O$ and a decrease in $O_3$, within the AMA. The negative $O_3$, caused by the inflow of lower tropospheric air with low $O_3$ mixing ratios, similar to the temperature anomalies, seems to be related to the anomalous horizontal flow as indicated by the streamlines of the horizontal wind components, regressed on the MIDX.

The regression coefficients of the MIDX on MMOD CCM temperatures, $H_2O$, and $O_3$ on the 360, 370, and 380 K isentropic surfaces (Fig. 12) indicate large areas with significant influences of the ASM, not only confined to the ASM region. On the 360 K level in the upper troposphere, a temperature increase with increasing ASM activity by about 5 K is located within the AMA over the TP coinciding with an increase in $H_2O$ by 65%. The TP acts as an elevated heat source in the mid–troposphere which makes a major contribution in forming the AMA, as shown by Liu et al. (2007). Convective events released over the TP

more often reach the tropopause than over the BoB (Fu et al., 2006). South to the AMA, the regression results on the 370 and 380 K isentropes indicate a cooling over the eastern Indian ocean and western Pacific warm pool, coinciding with decreasing $H_2O$ concentrations. These patterns of temperature anomalies, corresponding to strong ASM seasons, contrast the temperature pattern on 380 K for ERA-Interim (Fig. 11a, left), showing a positive temperature anomaly at the south-eastern edge of the AMA. However, the positive $H_2O$ anomalies within the AMA prevail on the 370 and 380 K isentropes.

The largest MMOD $O_3$ decrease of 36% on the 360 K isentropic level is located over the Eastern Mediterranean Sea, the western flank of the AMA, and a secondary $O_3$ decrease by 24% is located over north–eastern China at the eastern flank of the AMA. The pattern of the $O_3$ regression coefficients is slightly decreasing but persistent in height. The strongest MMOD temperature and $H_2O$ signals of the MIDX seem to be decoupled from that in $O_3$, which might be an indication for the more complex nature of $H_2O$, as it can change its phase during transport in regions of dehydration. With increasing height, on 370

and 380 K, the influence of the monsoon circulation on temperature within the AMA is decaying, whereas a temperature dipole





becomes obvious with decreasing temperatures over Indonesia and the western Pacific warm pool and increasing temperatures over the subtropical regions of the central Pacific during strong monsoon seasons. Increasing $H_2O$ concentrations are still present in the AMA at 370 and 380 K, although weaker than at 360 K, with 24% and 14% respectively.

## 5.2 Influence of ENSO

The ENSO influence on the zonal mean temperatures from in nine re-analyses datasets has been analysed by Mitchell et al. (2014), showing a warming in the tropical troposphere during ENSO warm events. Towards the tropical tropopause region this warming turns into a cooling. However, the main ENSO signature on temperatures in the tropics has strong longitudinal variation, mainly over the Pacific, which partly cancel each other when analysing zonal averages (Randel et al., 2000).

The Niño3.4 regression coefficients on ERA-Interim temperatures and $H_2O$ mixing ratios (Figure 11b) emphasize the zonal

asymmetric response with the typical horse shoe pattern of decreasing temperatures with increasing SSTs in the Niño3.4 region during an ENSO warm event. Simultaneously anticyclonic circulation cells to the northwest and southwest, and low $H_2O$ mixing ratios develop. This kind of pattern was first identified by Gill (1980) in an idealised model as the dynamic response to a heat source centred at the Equator. A similar pattern was also found in re–analysis fields in the TTL by Gettelman et al. (2001) and Zhou et al. (2001) to be caused by El Niño events. The Niño3.4 regression on ERA-Interim $O_3$ mixing ratios

shows an unexpected positive response in the central Pacific, whereas a negative signal would be more plausible, due to the outflow of $O_3$-poor above the eastward shifted convection during El Niño events.

We find the strongest impact on the CCM temperatures at the 370 K level with two centres located at 15°S and 15°N near 160°W, slightly decaying at 380 K and less pronounced at 360 K (Figure 13). The shift of convection towards the central pacific with less intense convection over Indonesia creates a dipole consisting of a cold and dry anomaly in the UTLS above

the central Pacific and a warm and wet anomaly above the western Pacific warm pool. These structures are quite the opposite to the regression patterns from the MIDX time series, indicating the higher probability of strong ASM seasons during La Niña events. The influence of ENSO warm events on the UTLS temperatures and $H_2O$ mixing ratios in the ASM region is less pronounced than the influence derived for the MIDX, and even though warming prevails for ENSO warm events above the ASM region, there is no significant increase in $H_2O$ mixing ratios north of 30°N. The higher $H_2O$ concentrations during ENSO

warm events span from the western Pacific towards Africa in the longitudinal direction and extend only to near south of 30°N in the ASM region. To the north over the TP the Niño3.4 regression coefficients on $H_2O$ indicate an insignificant decrease in mixing ratios. The highest percentage changes in $O_3$ concentrations during ENSO warm events are found at 360 K, again with a typical horse shoe-like pattern, showing a stronger impact in the northern subtropics. The influence of ENSO on $O_3$ seems to be weaker on the 370 and 380 K level.

The comparison of the Niño3.4 with the MIDX regression coefficients (Figure 12), reveals the similarity of the strong ASM cases with La Niña conditions which we suppose to be opposite the El Niño conditions as shown in Figure 13. This is supported by the results of the Niño3.4 regression with ERA-Interim data, showing also a double peak structure of decreasing temperatures in the subtropics of the central Pacific and increasing temperatures above the western Pacific warm pool during



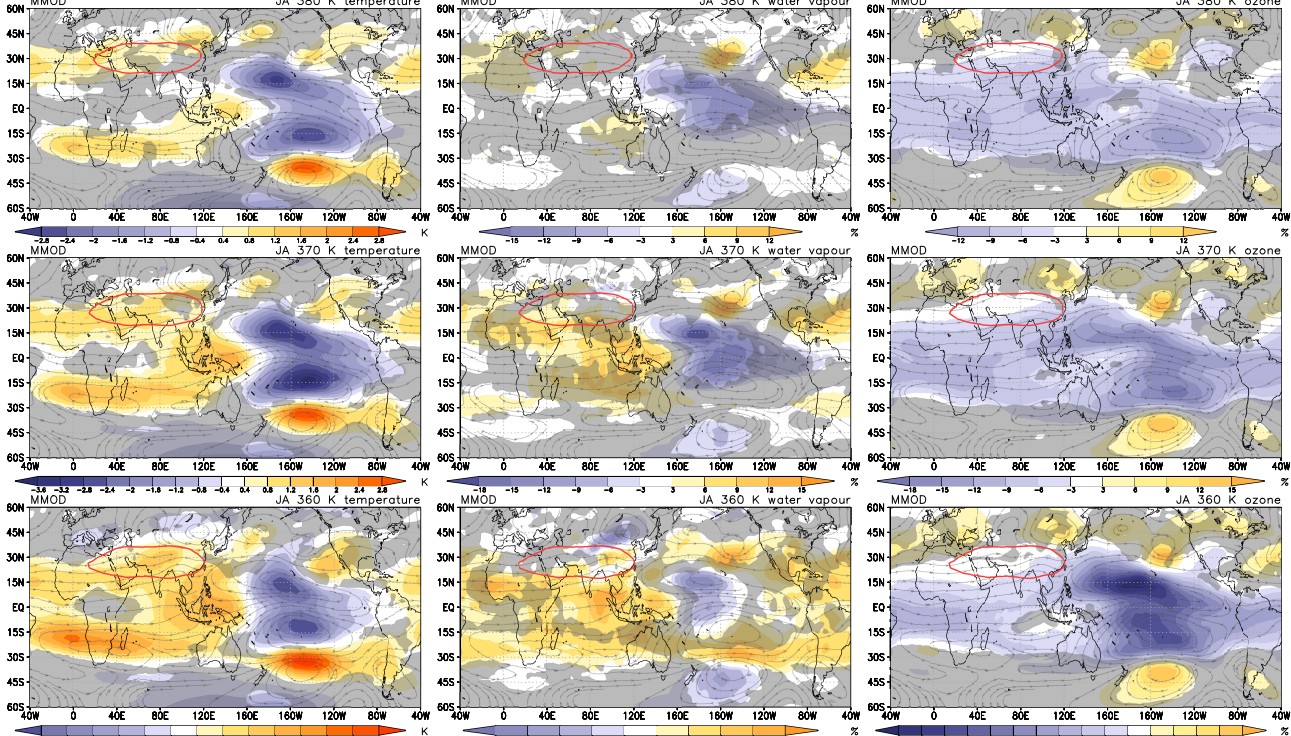

**Figure 13.** As Figure 12 but for the Niño3.4 index.

El Niño events. This temperature pattern is also reflected to a certain degree in the $H_2O$ mixing ratios, whereas the regression results for $O_3$ are not significant.

## 5.3 Influence of the Quasi Biennial Oscillation

The QBO is characterised by downward propagating vertical shear zones of the zonal wind. Westerly (easterly) shear zones are
5   creating positive (negative) temperature anomalies, according to the thermal wind balance. To maintain the QBO in temperatures, a secondary mean meridional circulation (MMC) arises with equatorial relative downwelling (upwelling) in westerly (easterly) shear zones of the zonal wind (Plumb and Bell, 1982). The QBO therefore modulates the strength of the prevailing upwelling in the equatorial lower and middle stratosphere. To ensure continuity, the air is forced to move upward (downward) in the subtropics when relative downwelling (upwelling) occurs at the Equator. Above and below the region of maximum rela-
10   tive downwelling, convergent and divergent motion close this QBO induced MMC. The QBO thereby affects the temperatures at the tropical tropopause (Zhou et al., 2001), and has the ability to modify the $H_2O$ concentrations entering the lower stratosphere (Giorgetta and Bengtsson, 1999). The QBO induced MMC is known to also affect $O_3$ transport with downwelling of $O_3$ rich air in a westerly shear zone, thereby generating a QBO in $O_3$ mixing ratios (e.g. Cordero et al., 1997; Logan et al., 2003).





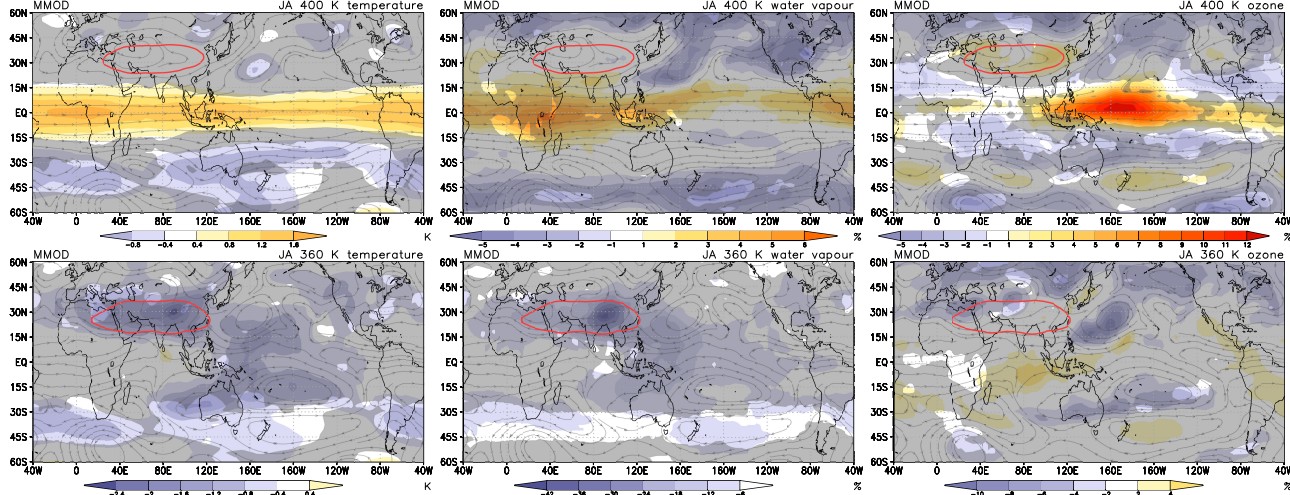

**Figure 14.** As Figure 12 but for the Quasi Biennial Oscillation on the isentropic levels 360 and 400 K (from bottom to top), including the CCMs: CCSRNIES, EMAC, EMAC-FUB, E39CA, SOCOL, UMUKCA-UCAM, and WACCM. The contour level for the 400 K Montgomery streamfunction is $3730 \times 10^2$ m$^2$ s$^{-2}$,

The QBO regression coefficients can be interpreted to represent the changes of the temperatures and the H$_2$O and O$_3$ mixing ratios forced by a mean amplitude between the easterly to westerly phase of the QBO. The modulation of the ERA-Interim 380 K temperatures by the QBO (Figure 11c) shows a significant increase in the inner tropics extending from the eastern Pacific to the Indian Ocean, and some regions of decreasing temperatures in the subtropics, consistent with the QBO induced

circulation changes. The changes in H$_2$O mixing ratios are consistent with the temperature changes, with increasing H$_2$O mixing ratios in the inner tropics where temperature increase is induced, and decreasing H$_2$O mixing ratios in the extra tropics. There is a remarkable gap in the QBO regression pattern for the temperatures and H$_2$O mixing ratios over the equatorial western Pacific, as probably some of the QBO induced variability might be described by the Niño3.4 index. The insignificant O$_3$ regression patterns shown for 380 K are very similar on the isentropic levels 360 and 370 K (not shown), and do not show

a plausible response to the QBO, which might be caused by assimilating total column ozone.

To emphasize the height dependence of the QBO influence, the QBO regression results on the CCM data that include a QBO in Figure 14 are shown for the 360 and 400 K isentropic surfaces. The temperatures increase close to the Equator at 400 K, as the adiabatic cooling is suppressed in the regions of less upwelling. The strongest increase of more than 1.5 K occurs near equatorial Africa. The temperature decreases in the subtropics in both hemispheres as a result of the QBO induced MMC that

creates anomalous rising of air.

The QBO regressions on the 400 K H$_2$O mixing ratios show the expected increase near the Equator, due to the anomalously high equatorial temperatures, and also an increase in 400 K O$_3$, due to anomalous downwelling of O$_3$ rich air. The QBO signal is more clear and consistent among the CCMs for temperature than for the O$_3$ and the H$_2$O mixing ratios. Similar to the ERA-Interim data, there is a deviation from the zonal nature of the QBO signal, as pronounced in temperatures, over the equatorial



central Pacific. The influence of the QBO diminishes at lower isentropic levels near the Equator, but subtropical latitudes still seem to be affected by the phase of the QBO, with a non–significant tendency for decreasing temperatures and $H_2O$ mixing ratios above the TP at 360 K.

## 6   Summary and Conclusions

The first part of the paper assessed the ability of a number of CCMVal-2 CCMs to reproduce the climatological $H_2O$ and $O_3$ distributions, the circulation patterns, and temperatures in the UTLS associated with the Asian summer monsoon (ASM). The climatological $O_3$ mixing ratios of the multi-model mean (MMOD) on isentropic surfaces (360–380 K) in the UTLS are in good agreement with MIPAS observations. There are, however, quite large differences with respect to the $H_2O$ mixing ratios, in particular a moist bias in high latitudes in the MMOD. Both tracers show considerable deviations of their extreme values within the Asian monsoon anticyclone (AMA) in the individual models, however the relative $O_3$ minimum and $H_2O$ maximum is captured by all models. Evident from the $H_2O$ mixing ratios on isentropic levels is also some missing confinement within the AMA in most models, but rather a strong north–eastward transport on the isentropes.

The horizontal and meridional circulation patterns, related to the AMA, are slightly weaker represented in the CCMs compared to ERA-Interim, except for the meridional winds at the eastern flank of the AMA. The CCMs are warmer in tropical and subtropical latitudes during JA at the 360 K surface, compared to ERA-Interim, but get colder at 370, and 380 K. However, CCMs show a weaker ASM induced temperature anomaly than ERA-Interim. We have to stress that individual models deviate from this overall behaviour of the MMOD.

The second part of the paper identified factors, influencing the interannual variability of temperatures, $H_2O$, and $O_3$ in the UTLS. We performed a multiple linear regression analysis, including a derived monsoon circulation index (MIDX), the Niño3.4 index, and the QBO. By definition, the MIDX is a measure of strength of the ASM in the upper troposphere at 150 hPa. Regressing the MIDX on ERA-Interim temperatures suggests for strong ASM seasons an upper tropospheric warming southeast of the AMA and two separated areas of cooling, one in the western part of the AMA and one to the northeast of the AMA. Regressing the MIDX on NOAA OLR data indicates decreases of OLR in many regions during July and August, thus indicating an increase in deep convection south of the AMA core. Regions where MIDX correlates to OLR differences or temperature changes are largely dislocated. While the OLR differences reflect the areas of strongest convection in the ASM, the subtropical dipole of temperature changes with centres over Iran and north-eastern China reflects the dynamical response to the diabatic convective heating. The corresponding regression patterns for $H_2O$ and $O_3$ mixing ratios emphasize the ability of strong ASM seasons to moisten the air within the AMA, despite a colder tropopause region, whereas $O_3$ mixing ratios tend to be lower. Hence, we can not support the results of Randel et al. (2015) that a weaker ASM season induces increased $H_2O$ mixing ratios within the AMA. These differences may however be due to the different approaches used in both studies.

The regressions of the MIDX on the temperatures of the CCMs indicate that the CCMs do not capture the warming to the south-east, as shown for the ERA-Interim temperatures. Instead UTLS temperatures at 360 K increase with monsoon intensity in the ASM region centred above the Tibetan Plateau (TP), which enhances the seasonal high climatological temperatures over





the elevated TP. The anomalous warming coincides with increased $H_2O$ and decreased $O_3$ mixing ratios. At 370 and 380 K increased $H_2O$ is still present above the TP but the region of increased $H_2O$ is extending to the north–east, confirming the potential of the ASM in moistening the lowermost stratosphere at higher latitudes during boreal summer (e.g. Rosenlof et al., 1997; Dethof et al., 1999; Ploeger et al., 2013). The pattern of decreased $O_3$ concentrations during strong ASM seasons is quite

persistent on all three isentropic levels, that are located at the western and northeastern edge of the AMA.

It is well known that ENSO and ASM strength are not independent, e.g. after an ENSO warm event the strength of the ASM is often reduced (Ju and Slingo, 1995). In our study some patterns of the regression coefficients of the Niño3.4 index mirror, as expected, the MIDX regression patterns near the Equator between 15°S–15°N, reflecting the shift of the most intense convection to the central Pacific during ENSO warm events. We note that the ENSO modulation of the ASM is non–negligible

and that non–linear interactions may exist, that are likely to complicate the unambiguous detection of the MIDX signal.

The results from the regression of the QBO time series on temperature, $H_2O$ and $O_3$ mixing ratios confirm the expected modulation of the transport in the UTLS by the QBO induced meridional circulation. The CCMs can reproduce the QBO induced temperature anomalies on the 400 K isentropic surface, i.e. a warming near the Equator and cooling in the subtropics, as well as the $O_3$ transport anomalies, and to a certain degree the $H_2O$ anomalies that are related to the temperature changes.

The results of the CCMs during strong ASM seasons, confirm the importance of the TP and the southern slope of the Himalayas for the $H_2O$ transport to the UTLS above the ASM region (e.g. Fu et al., 2006; Wright et al., 2011; Bergman et al., 2013). The coinciding positive temperature anomalies suggest transport through this region, rather than transport through the southern flank of the AMA, where temperatures are lower during strong ASM seasons. Results from CCMs and re–analyses further indicate that $H_2O$ is transported towards higher latitudes on isentropic levels as suggested previously by Dethof et al.

(1999) and Ploeger et al. (2013), rather than fed into the tropical UTLS to contribute to the $H_2O$ tape recorder seasonality, as proposed by other studies (e.g. Bannister et al., 2004; Gettelman et al., 2004).

**Appendix A: Estimating the significance of averaged regression coefficients**

The statistical significance of the estimated fit parameters is tested with a two–tailed Student $t$-test of the null hypothesis $H_0 : \beta_j = 0$ with the alternative hypothesis $H_1 : \beta_j \neq 0$. The regression parameters of the individual CCMs are averaged to

get a combined response of all CCMs as a multi model average. To decide about the significance of the combined regression parameters, the weighted $Z$-test (see Whitlock, 2005, and references therein) is used which combines the p–values from the MLR of the individual ($i = 1, \ldots, k$) CCMs.

$$Z_w = \frac{\sum_{i=1}^{k} w_i Z_i}{\sqrt{\sum_{i=1}^{k} w_i^2}} \tag{A1}$$

The weighted $Z$–transform ($Z_w$) is created by applying the weights $w_i = se^{-2}$ (inverse of the squared standard error of the

regression coefficients) to the standard normal deviates $Z_i$, that are created from the $p_i$–values of the $t$-test on the individual regression coefficients as $Z_i = \Phi^{-1}(p_i)$, with $\Phi^{-1}$ the inverse normal cumulative distribution function. The resulting $Z_w$





indicate that, at least for one of the CCMs, $H_0 : \beta_j = 0$ is rejected, when the 97.5% quantile is reached, corresponding to a $Z_w \leq -1.96$.

### Appendix B:  Autocorrelations of the residuals

Inherent to all kinds of meteorological time series data is their tendency to exhibit autocorrelations. The independence of the individual data values is therefore often not fulfilled. This has the serious consequence of underestimated uncertainties, derived from the multiple linear regression model indicating significance for insignificant results. To avoid this the residuals $(\varepsilon(t), t = 1, n)$ are tested for autocorrelations with a second order autoregressive model, $\varepsilon(t) = \rho_1 \varepsilon(t-1) + \rho_2 \varepsilon(t-2) + a(t)$, after the regression model has run for a first time. The autoregressive parameters $\rho_1$ and $\rho_2$ are then used to transform the model according to Tiao et al. (1990), e.g. $y'(t) = y(t) - \rho_1 y(t-1) - \rho_2 y(t-2)$. The transformation is applied in the same way to the time series of the response variable and the time series of the basis functions. Whereas the uncertainties of the original response and basis functions are set to one for the first run of the least square regression, they are calculated according to Box and Jenkins (1970) for the second run with

$$\sigma_t = \sqrt{\left(\frac{1-\rho_2}{1+\rho_2}\right)\frac{\sigma_\varepsilon^2}{[(1-\rho_2)^2 - \rho_1^2]}} \tag{B1}$$

where $\sigma_\varepsilon^2$ is the variance of the residuals.

*Acknowledgements.* We acknowledge the modelling groups for making their simulations available for this analysis, the Chemistry-Climate Model Validation (CCMVal) Activity for WCRP's (World Climate Research Programme) SPARC (Stratospheric Processes and their Role in Climate) project for organizing and coordinating the model data analysis activity, and the British Atmospheric Data Center (BADC) for collecting and archiving the CCMVal model output. European Centre for Medium-Range Weather Forecasts (ECMWF) ERA-Interim data used in this study have been obtained from the ECMWF data server. Interpolated OLR data provided by the NOAA/OAR/ESRL PSD, Boulder, Colorado, USA, from their Web site at http://www.esrl.noaa.gov/psd/. MK was supported by the European Community within the StratoClim project (grant no. 603557).



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
