# Peer review of "Interannual variability of the boreal summer tropical UTLS in observations and CCMVal-2 simulations"

_Atmospheric Chemistry and Physics, 2015_

## Referee Comment (RC1) · Anonymous Referee #1 · 15 Feb 2016

The paper presents an assessment of the representation of the Asian Monsoon Anticyclone (AMA) in the UTLS in the CCMVal 2 models. The ERA-Interim reanalysis and the MIPAS satellite data are used to compare the dynamical variables and the water vapour and ozone concentrations. The manuscript provides new insights on the abilities and limitations of the models in representing the AMA structure and tracer distribution. The paper is nicely written and the methodology is accurately explained. I recommend publication in ACP after the following minor comments are addressed.

- P5 L14: This is the first time the terminology JA is used in the main text, explain what it means.

- Figure 1. In the top panel, showing the winter values of the MIDX is unnecessary and masks the interannual variability in the time series. I would suggest showing the sum-
[Figure]

mer values alone in this Figure, in particular the JA indices used for the regressions.

- P8 L12-13: Why do you need to multiply the fit parameters by some factor? Also, in Eq. (2) some regression terms are included that are not shown in the paper (volcanoes, trend, solar). Is there a reason to include them?

- Figure 3 caption: divergence-free zonal wind anomalies

- P12 L4: Do you have an idea of why the models have higher water vapour values?

- In the main text Figure 7 is discussed before Figure 6. I suggest that you switch the order of the Figures.

- P14 L3-4: This sentence is not clear, what do you mean?

- P15 L6: remove comma

- P16 L4-5: What is the reason to show the 120E-160E?

- P16 L24-27: This sentence is too long, I suggest adding a parenthesis in: (annual average ... monsoon circulation)

- Figure 10 and related discussion (P2 L11-14): The main difference in OLR pattern with the Randel et al. (2015) results is in South-East China, where their results show reduced convection over a broad area, and Figure 10 in this paper does not show any significant anomaly. I do not see strong differences in the Bay of Bengal. Also, Randel et al. (2015) argue that anomalously cold temperatures associated with strong convection lead to stronger dehydration reducing water vapour in the subtropical UTLS. Do you propose an alternative mechanism? This should be clarified when contrasting your results to those of the mentioned paper. These comments refer also to the discussion on P25 L20-30.

- P21 L4: The positive anomaly is centred on an island in front of the Vietnam coast called Hanai.

---

## Referee Comment (RC2) · Anonymous Referee #2 · 18 Feb 2016

General comments:

This paper contains some interesting analyses regarding transport dynamics associated with the Asian summer monsoon that could potentially be important scientific contributions and worthy of publication in Atmospheric Chemistry and Physics. However, as it stands, the paper has severe weaknesses that make it unpublishable in its current form. The most glaring problems are that it is poorly organized, lacks focus, and the logic of their analysis is often lost in descriptive details that are not relevant to the main points of the paper. An example of the lack of focus concerns Sec. 4 that discusses the 'assessment' of atmospheric models from CCMVal to reproduce temperature, water vapor and ozone distributions. This section is awash with details and confusing analysis that neither produces a finding significant enough to state in the abstract nor one that is (apparently) used in subsequent analysis.

There are also problems with the logic of the arguments themselves – although the organizational problems of the paper make it difficult to properly assess the logic of these arguments. Consider the contention that the Tibetan plateau is the primary source of water vapor for the monsoon anti-cyclone. The sole evidence for that seems to be a peak in the correlation between temperature variability and their monsoon index found in the atmospheric models over the Tibetan plateau (at the 360 K potential temperature surface). Not only does this argument fail to show any dynamical relationship to water vapor variability, but the reliability of temperature signal is questionable; in comparisons to ERA-interim, the authors do not show values at 360 K, but they do show values at 380 K that are not in good agreement. A second example concerns the contention the anti-cyclone transports water into the mid-latitude stratosphere, but not into the tropical pipe. It seems that the sole evidence for this is the large water vapor mixing ratios found to the north of the anti-cyclone, but not to the south. Such a water vapor distribution could certainly arise for the reason that the authors contend, but they provide no evidence for a dynamical link that shows that the high water vapor concentrations are due, at least in part, to transport by the anti-cyclone. Furthermore, the results from Sec. 4 show large discrepancies between water vapor distributions from the CCMVal models and satellite observations.

I suggest that the paper be rejected but that the authors be encouraged to resubmit after a thorough overhaul that provides clear logical arguments backed up with strong evidence. For the revision, I suggest that the authors choose some small number (e.g., 3-5) of important points, choose the results that best illustrate those points and how uncertain they are, and then rewrite Secs. 4 and 5 accordingly. In particular, there are important aspects of the circulation that are not well reproduced by the CCMs (for example, water vapor distributions) that could easily undermine the results. A more careful and logical discussion will help determine just how strong of a case the authors actually have.

Specific Comments:

Many of the figures are crowded into too small of a space and not well labeled, making them difficult to understand. Please add explicit labels on the figures that help distinguish the different panels. For example: Fig. 1 should label the top panel MIDX and the bottom panel WIDX and Nino 3.4, Fig. 2 should label the top panels as 150 hPa stream function and the bottom panels velocity potential, and so on. Also, consider dividing the individual panels in a way that does not crowd them into such small spaces. Perhaps some of the panels can be left out of the paper.

Page 1, lines 4-7: State briefly what the CCM assessment is.

P. 1 L. 15-16: Be more clear: what is meant by 'consistent'? Weaker than what?

P. 7 L. 11-12: It is over-simplified to state that the 3 terms in equation (1) 'represent' the Hadley , Walker, and Monsoon circulations. That is, each term contains more than just those circulation features. It would be, for example, better to say something like 'Chi-star-prime is influenced by the monsoon circulation'.

P. 7, L. 17: Explain why you add an artificial seasonal cycle to MIDX (by changing from using maximum values of $\chi$ to minimum values). Given the strong seasonality of the monsoon, it should be possible to find an index that has a strong seasonal cycle without any artificial inflation.

Figures 3 and 4: The discussion of these figures is particularly chaotic and confusing.

P. 21 L. 4-5: Please clarify this discussion. It seems to me that Fig. 10a indicates positive regressions for MIDX onto OLR over BoB, Myanmar, and Taiwan. Doesn't that indicate weaker convection over these regions – instead of stronger as you state? Or do you mean to say 'we also get a decrease in convective activity over the BoB . . .'

Appendices: The Appendices are too terse to be useful. The autocorrelations discussed in Appendix B are not referred to in the main text (except at the end of Sec. 3 which merely states that autocorrelations were treated) and should be removed unless a more articulate discussion of how the autocorrelations affect the analysis is provided.

Appendix A is also not necessary. You could simply mention that the criteria for significance are derived from the Z-test and refer the reader to Stouffer et al. (1949) and Whitlock (2005) for details.

Technical details:

Fig. 1 caption: Add the term 'WIDX' to the description. For example, 'bottom: index for the Walker circulation (WIDX; solid)'

P. 6 L. 4: Regarding 'graduate'. Do you intend this word to mean 'to make more gradual'? If so, this is an awkward use – if not, it's difficult to understand the meaning of the phrase. It would be better to use a different word.

P. 6, L. 8: Change 'indication for the' to 'indication of the '.

P. 6 L. 10: No comma after 'model'.

P. 7 L. 5: Change 'allows to express' to 'allows us to express' or some other grammatically correct wording.

P. 7 L. 24: The clause 'whereas the nino3.4 . . . variability' is a non sequitur – it implies contrasting behavior but no source of the contrast is given. Perhaps you intend to say something like 'the regression onto WIDX emphasizes the west Pacific circulation response to inter-annual SST variations whereas regression onto nino3.4 describes the larger (scale) response to ENSO variability.'

P. 10 L3: Change relative to 'relatively'. Also you should state what you are comparing to when you say it is relatively small (i.e., relative to what?).

P. 12 L. 6: Add (e.g., in parentheses) that the heating rates are displayed with red lines.

P. 21 L. 13: Change 'round' to 'around'

P. 21 L. 25: Change 'South to the AMA' to 'South of the AMA'.

P. 22 L. 5: Remove 'in' from 'temperature from in nine re-analysis datasets'.

---

## Referee Comment (RC3) · Anonymous Referee #3 · 17 Mar 2016

**General:**

Validation of the chemistry climate models CCMs) with respect to their abilities to represent the Asian summer monsoon (ASM), especially the Asian monsoon anticyclone (AMA) is an important task for the atmospheric community. The paper uses MIPAS and ERA-Interim data to validate such CCMs; the comprehensive analysis is clear and well presented. In the second part, the interannual variability of the ASM/AMA system is considered. However, there are some major points which need a more detailed discussion.

**Major points:**

1. Fig 3 and 4
   Both figures show the results relative to the tropopause pressure that is certainly a good idea. You write that in order to account for differences among the CCMs in the location of AMA, the mean anomaly averaged over 30 degree was "centered where the 150 hPA eastward directed divergence free zonal wind maximizes". However, I would like to see such differences in the model representation and would recommend to use a much simple averaging over 120-160E. Maybe you can make two figures for this (you do something similar in Fig 8). Furthermore, the most important information shown in Fig 3/4 are for me temperature anomalies (rather than wind anomalies) which are extremely difficult to read. A compromise could be to show wind anomalies in the absolute range 120-160E and temperature anomalies by using the relative coordinate defined by the wind maximum (and only to mention in the text that such "shifted" wind patterns are very similar for ERA-Interim and the MMOD analysis) .

2. Fig 10 and the discrepancy with Randel et al 2015
   This is a very interesting and important point. However, a simple explanation referring to "different approach" is not enough for me. You can certainly repeat the Randel's procedure by using ERA-Interim H2O (instead of MLS like in Randel et al.). If you get a similar picture ("more convection makes a dry anomaly") than is your statement ("different approach") correct. Otherwise, without such a test you have a "confusing result" if compared with the published work of Randel et al 2015. Furthermore, the paper is in my opinion too long. I would recommend to publish two parts: (1) validation with MLS/ERA-Interim and (2) Iterannual variability. But, that is your decision.

**Minor points:**

1. General
   In almost all your figures you use a matrix of sub-panels. It would be easier to read such figures if you would denote every row and every column separately. E.g. Fig 5/6 $\theta = 360$ 370, 380 K for the rows and MIPAS/MMOD for the columns.

2. P1, abstract, L14-15
   please mention "zonally asymmetric ENSO response versus zonally symmetric QBO modulation"

3. P1, L24
   first "wave-driven" forcing, followed by heat transport from the tropics to the high latitudes and, finally slow ascent due to radiative heating - please reformulate

4. P 2, L 20-25
   To discuss the importance of the Tibetan Plateau you should also mention the Boos and Kuang, Nature 2010 paper stating that for the formation of the Asian monsoon circulation pattern orography is the most important factor and the impact of sensible heat (Tibetan Plateau) is rather a second order effect

5. P3 L 10-13
   Maybe you should discus it more carefully: the core of the anticyclone is rather in the extratropics than in the tropics. Furthermore, the anticyclone itself acts more as an isentropic blower. Inside of the anticyclone the the tropospheric pollution are trapped and probably transported into the TTL (Randel et al., Science, 2010). Outside of the anticyclone a strong in-mixing of stratospheric signatures into the TTL happens (see related paper from Konopka et al and Ploeger et al)

6. P3 L14
   ...(QBO) or the "internal variability of the ASM iteslf".

7. P4, L26
   "aspects of the climatological state are compared with" - which aspects, please reformulate

8. P6, caption Fig 1
   please use the abbreviation WIDX
9. P6, L5
   "graduate" - I am not sure that this is a right word. Maybe "mask" or "suppress"

10. P7, L5
    Explain the vector $k$

11. P9 L2
    Use the notation $\psi$ for the divergence-free part of the flow. Same for $\chi$ (which were defined in the previous section).

12. P12 Fig 5
    The enhanced signatures of H2O north of 30N seem to propagate eastward mainly by planetary waves as described by Ploeger et al. Maybe you would like to include some comments about this point

13. P14 Fig 7
    There are much lower temperatures at 380 K for MMOD than for ERA. On the other side MMOD are moister compared with MIPAS. You should comment this point

14. P14 L9
    "O3 in the UTLS can better serve as a passive tracer..." - maybe you can make this point earlier, e.g. as you introduce O3 into your discussion

15. P15 Fig 8
    After the major point 1 was included, Fig 8 would be easier to understand

16. P15 last sentence and P16 first sentence
    This feature was discussed in literature as in-mixing, see Konopka et al 2009, 2010, Ploeger et al 2012. Maybe you would like to include these references into your discussion

17. P17 caption of Fig 7 (and Fig 2)
    You introduced the decomposition given by the eq (1) but you do not use the introduced notation. Please state it explicitly if you show $\psi$, $\chi$, $\chi^*$, etc.

18. P17 L13
    For me MIDX is a more direct measure of the anticyclone rather than of the whole ASM system

19. P18 Fig 10 and the discrepancy with Randel et al 2015
    see major point 2

20. P19 Fig 11
    I think, you use the ERA-Interim related results too strong as a benchmark for the following investigations. Whereas ERA-Interim temperatures and probably H2O are good enough for your study, ERA-Interim ozone around and below the tropical tropopause is probably not good enough for that (mainly because only O3 column is constrained by sat elite observations as described in Dragoni et al., 2011) . In the following you describe large differences in ozone between multi-model average of the CCMs and the ERA-Interim. I would recommend to exclude completely the ERA-Interim ozone.

21. P20 L7
    "As MIDX is a direct measure of the strength in upwelling" - for me MIDX is a direct measure of the (divergence) of the anticyclone, please re-formulate

22. P20 L9
    ...or have increased H2O or less O3

23. P20 L10
    ...or decreased H2O or higher O3

24. P21 L17
    "The negative O3 caused...." - I do not understand your explanation. Negative O3 anomaly means a stronger tropospheric influence (more upwelling) that is in agreement with the positive H2O anomaly. Please clarify

25. P22 L15
    "unexpected positive response" - see comments above to ERA-Interim ozone

26. P25 L23
    "many regions" - please list these regions

27. P26 L17
    ...suggest transport of H2O through this region

---

## Author Comment (AC1) · 13 May 2016

**Response to referee #1.**

We thank the referee for the constructive comments that helped to improve the paper. (Referee comments are emphasized in *italics*.)

*The paper presents an assessment of the representation of the Asian Monsoon Anticyclone (AMA) in the UTLS in the CCM-Val 2 models. The ERA-Interim reanalysis and the MIPAS satellite data are used to compare the dynamical variables and the water vapour and ozone concentrations. The manuscript provides new insights on the abilities and limitations of the models in representing the AMA structure and tracer distribution. The paper is nicely written and the methodology is accurately explained. I recommend publication in ACP after the following minor comments are addressed.*

We thank the referee for this encouraging appraisal of our manuscript. All further questions/comments are answered/annotated in the following and the manuscript is changed accordingly.

*- P5 L14: This is the first time the terminology JA is used in the main text, explain what it means.*

The terminology JA (July/August) is already explained on page 3/line 26.

*- Figure 1. In the top panel, showing the winter values of the MIDX is unnecessary and masks the interannual variability in the time series. I would suggest showing the summer values alone in this Figure, in particular the JA indices used for the regressions.*

The updated version of the figure is now showing the time series of the JA average of the MIDX, the Nino3.4, and the QBO, as these are the basis functions used in the multiple linear regression model.

*- P8 L12-13: Why do you need to multiply the fit parameters by some factor? Also, in Eq. (2) some regression terms are included that are not shown in the paper (volcanoes, trend, solar). Is there a reason to include them?*

The regression coefficients reflect the change of a quantity, for e.g. JA, by one unit of the respective basis function. To get an estimate of the changes expected by a typical amplitude of the basis function, the regression coefficient is multiplied by a mean amplitude.
A regression term for the trend should be part of any multiple linear regression model. The regression terms for volcanoes and the solar flux are included in the regression model to capture the interannual variability that potentially emerges by these sources of natural variability. The additional regression terms are necessary to better isolate the signals of the ASM, ENSO and the QBO. The results of the additional regression terms are not discussed in the paper to focus on the most relevant factors.

*- Figure 3 caption: divergence-free zonal wind anomalies*

Figure 3 is removed from the manuscript.

*- P12 L4: Do you have an idea of why the models have higher water vapour values?*

These large water vapour mixing ratios can for some CCMs be related to a warm bias in the respective region. But in general this might be too simple as an explanation. Area averaged 360 K ERA-Interim temperature are higher than for the MMOD but MIPAS water vapour maxima (and also ERA-Interim, not shown) in the respective region are lower than the water vapour maximum of the MMOD. We included a sentence about this in the revised manuscript.

*- In the main text Figure 7 is discussed before Figure 6. I suggest that you switch the order of the Figures.*

The order of the figures is switched in the revised manuscript.

*- P14 L3-4: This sentence is not clear, what do you mean?*

Between 360 and 380 K dehydration reduces the MIPAS peak values of the H2O mixing ratios in the ASM region to 7 ppmv at 380 K but this maximum is not as pronounced as at 370 or 360 K.
Additional explanations are now included in the revised manuscript.

*- P15 L6: remove comma*

Done.

*- P16 L4-5: What is the reason to show the 120E-160E?*

The region is characterized by a meridional circulation directed towards the Equator, which is a feature of the eastern edge of the AMA. In addition uplift is prevailing, as indicated by the upward arrows. It is also a region where tracers might have been drawn out of the AMA core region.
The choice of this region is now better motivated in the revised manuscript.

*- P16 L24-27: This sentence is too long, I suggest adding a parenthesis in: (annual average ... monsoon circulation)*

To increase the readability, the parenthesis are included in the sentence.

*- Figure 10 and related discussion (P2 L11-14): The main difference in OLR pattern with the Randel et al. (2015) results is in South-East China, where their results show reduced convection over a broad area, and Figure 10 in this paper does not show any significant anomaly. I do not see strong differences in the Bay of Bengal.*

The estimation of the strength of the Monsoon in Randel et al. (2015) is based on the OLR anomalies above the southern Tibetan Plateau. Regarding the OLR anomalies above the Indian subcontinent it would be also possible to come to a different estimate.

*. . . Also, Randel et al. (2015) argue that anomalously cold temperatures associated with strong convection lead to stronger dehydration reducing water vapour in the subtropical UTLS. Do you propose an alternative mechanism? This should be clarified when contrasting your results to those of the mentioned paper. These comments refer also to the discussion on P25 L20-30.*

The regression results of the MIDX on temperature also show two regions of lower temperatures associated with stronger ASM (i.e. stronger convection). But rather than located directly above the region of most intense convective activity, as shown by Randel et al. (2015), the regions of cooling are shifted to the north-east and north of the region of convection well confined to the region of anticyclonic circulation anomalies associated with a strengthening of the AMA. A study by Bergman et al. (2013) identified a vertical conduit, located over the southern part of the TP and the southern slope of the Himalayas, where trajectories seem to be concentrated regionally. At lower levels and above this conduit the trajectories spread out to a larger area. If we apply this view to our results of the MIDX regression on ERA-Interim data, it could be possible that the H2O is preferably transported upward through the relatively warm region at the south-eastern edge of the AMA.

*- P21 L4: The positive anomaly is centred on an island in front of the Vietnam coast called Hanai.*

The text has changed accordingly.

**References**

Bergman, J. W., Fierli, F., Jensen, E. J., Honomichl, S., and Pan, L. L.: Boundary layer sources for the Asian anticyclone: Regional contributions to a vertical conduit, Journal of Geophysical Research: Atmospheres, 118, 2560–2575, doi:10.1002/jgrd.50142, 2013.

Randel, W. J., Zhang, K., and Fu, R.: What controls stratospheric water vapor in the NH summer monsoon regions?, J. Geophys. Res.: Atmos., doi:10.1002/2015JD023622, 2015.

5

---

## Author Comment (AC2) · 13 May 2016

**Response to referee #2.**

We thank the anonymous referee for the constructive comments that helped to improve the paper. All further questions/comments are answered/annotated in the following, the manuscript is changed accordingly. (Referee comments are emphasized in *italics*.)

*This paper contains some interesting analyses regarding transport dynamics associated with the Asian summer monsoon that could potentially be important scientific contributions and worthy of publication in Atmospheric Chemistry and Physics. However, as it stands, the paper has severe weaknesses that make it unpublishable in its current form. The most glaring problems are that it is poorly organized, lacks focus, and the logic of their analysis is often lost in descriptive details that are*

10 *not relevant to the main points of the paper. An example of the lack of focus concerns Sec. 4 that discusses the 'assessment' of atmospheric models from CCMVal to reproduce temperature, water vapor and ozone distributions. This section is awash with details and confusing analysis that neither produces a finding significant enough to state in the abstract nor one that is (apparently) used in subsequent analysis.*

15 Section 4 is intended to summarize the main climatological features of the Asian summer monsoon (ASM) circulation, water vapour and ozone mixing ratios as simulated by the CCMs in comparison to the ERA-Interim re-analyses or MIPAS. This task can be seen as an addition to the CCMVal report (SPARC CCMVal, 2010). We think that it is important to show the climatological analyses before showing the results of the multiple linear regression. We tightened Section 4 by removing Figures 3 and 4 from the manuscript and revised the text as proposed by the reviewer.

20

*There are also problems with the logic of the arguments themselves – although the organizational problems of the paper make it difficult to properly assess the logic of these arguments. Consider the contention that the Tibetan plateau is the primary source of water vapor for the monsoon anti-cyclone. The sole evidence for that seems to be a peak in the correlation between temperature variability and their monsoon index found in the atmospheric models over the Tibetan plateau (at the 360 K po-*

25 *tential temperature surface). Not only does this argument fail to show any dynamical relationship to water vapor variability, but the reliability of temperature signal is questionable; in comparisons to ERA-interim, the authors do not show values at 360 K, but they do show values at 380 K that are not in good agreement.*

At the 360 K isentropic level there is a strong and significant increase in ERA-Interim water vapour ($H_2O$) mixing ratios

30 and temperatures above the Tibetan Plateau (TP) with increasing monsoon circulation index (MIDX). To support our argument, the ERA-Interim MIDX regression patterns at 360 K are included in the revised manuscript. Temperature and water vapour variability are closely linked in the respective region. To show this we composited the July–August 360 K ERA-Interim temperatures and $H_2O$ mixing ratios (1979–2013) according to wet and dry extremes in a region above the TP defined as 70–100°E/30–40°N (Fig. 1, also included in the supplement as Fig. S6). There is a clear indication that months with $H_2O$ mixing

35 ratio anomalies enhancing by more than one standard deviation above the TP are connected with higher temperatures at 360 K above the TP and vice versa for the dry composite with months where the $H_2O$ mixing ratio anomalies are decreased by more than one standard deviation.

*... the authors do not show values at 360 K, but they do show values at 380 K that are not in good agreement.*

40

This is deliberate. We highlight that the MIDX regression pattern of the 380 K temperature is different from that at 360 K, as it shows the two areas of decreasing temperatures connected with the anticyclonic circulation patterns at the western edge and to the north-east of the anticyclone.

45 *A second example concerns the contention the anti-cyclone transports water into the mid-latitude stratosphere, but not into the tropical pipe. It seems that the sole evidence for this is the large water vapor mixing ratios found to the north of the anti-cyclone, but not to the south. Such a water vapor distribution could certainly arise for the reason that the authors contend, but they provide no evidence for a dynamical link that shows that the high water vapor concentrations are due, at least in part, to transport by the anti-cyclone. Furthermore, the results from Sec. 4 show large discrepancies between water vapor distributions*

*from the CCMVal models and satellite observations.*

There are several studies analysing isentropic water vapour transport (e.g. Dethof et al., 1999; Ploeger et al., 2013) that come to the conclusion that the ASM region is a significant moisture source for the lower stratosphere in northern high latitudes. The process responsible for this water vapour transport is described by Dethof et al. (1999) as an interaction of synoptic disturbances with the Asian monsoon anticyclone (AMA) that pull filaments of tropospheric air from the northern flank of the AMA. Our analyses are based on the monthly mean data base available for the CCMs participating in CCMVal-2, therefore it is not possible to study processes occurring on a much smaller time scale. On the other hand we have CCM data spanning 45 years which is much longer than the time series used for most studies related to the ASM.

*I suggest that the paper be rejected but that the authors be encouraged to resubmit after a thorough overhaul that provides clear logical arguments backed up with strong evidence. For the revision, I suggest that the authors choose some small number (e.g., 3-5) of important points, choose the results that best illustrate those points and how uncertain they are, and then rewrite Secs. 4 and 5 accordingly. In particular, there are important aspects of the circulation that are not well reproduced by the CCMs (for example, water vapor distributions) that could easily undermine the results. A more careful and logical discussion will help determine just how strong of a case the authors actually have.*

*Specific Comments: Many of the figures are crowded into too small of a space and not well labeled, making them difficult to understand. Please add explicit labels on the figures that help distinguish the different panels. For example: Fig. 1 should label the top panel MIDX and the bottom panel WIDX and Nino 3.4, Fig. 2 should label the top panels as 150 hPa stream function and the bottom panels velocity potential, and so on. Also, consider dividing the individual panels in a way that does not crowd them into such small spaces. Perhaps some of the panels can be left out of the paper.*

We reduced the number of figures in the revised manuscript to focus more on the interannual variability of the AMS. Remaining figures are now more clearly labelled.

*Page 1, lines 4-7: State briefly what the CCM assessment is.*

The word 'assessment' might be a misnomer in this context and therefore we replaced it by 'comparison'. As the reviewer correctly states, an assessment is a much deeper evaluation of CCMs, which also includes the rating of individual models.

*P. 1 L. 15-16: Be more clear: what is meant by 'consistent'? Weaker than what?*

The QBO regression results in water vapour and ozone are consistent in the sense that they are in agreement with the understanding of the QBO modulation. The downward propagating QBO west-phase is generating an anomalous meridional overturning circulation that is directed downward near the Equator and directed upward in the subtropics. The resulting temperature anomaly is positive, thereby leading to increasing water vapour mixing ratios. The reduced upwelling near the Equator is leading to higher ozone mixing ratio. The term 'weaker' was intended to evaluate the QBO regression patterns for water vapour and ozone in comparison to the QBO regression pattern for temperature. The statement is rephrased to be more obvious.

*P. 7 L. 11-12: It is over-simplified to state that the 3 terms in equation (1) 'represent' the Hadley, Walker, and Monsoon circulations. That is, each term contains more than just those circulation features. It would be, for example, better to say something like 'Chi-star-prime is influenced by the monsoon circulation'.*

The revised manuscript now includes a statement about the limitation of the Tanaka et al. (2004) method to separate tropical circulations.

*P. 7, L. 17: Explain why you add an artificial seasonal cycle to MIDX (by changing from using maximum values of Chi to minimum values). Given the strong seasonality of the monsoon, it should be possible to find an index that has a strong seasonal*

*cycle without any artificial inflation.*

The seasonal cycle is not artificially added to the MIDX time series, as it is a consequence of the reversal of $\chi^{*\prime}(t, x, y)$ over south-east Asia from being positive to negative during the months from October to April. To reflect this reversal in sign, the method of Tanaka et al. (2004) defines the value of the MIDX to be the maximum in $\chi^{*\prime}(t, x, y)$ in a region limited to south-east Asia from May to September and to be the minimum during the remainder of the year. As we are concentrating on the interannual variability of the ASM during July/August (JA), we use the JA average of the MIDX as a basis function in the multiple linear regression model that does not include the seasonal cycle.

*Figures 3 and 4: The discussion of these figures is particularly chaotic and confusing.*

The figures 3 and 4 are now removed from the manuscript.

*P. 21 L. 4-5: Please clarify this discussion. It seems to me that Fig. 10a indicates positive regressions for MIDX onto OLR over BoB, Myanmar, and Taiwan. Doesn't that indicate weaker convection over these regions – instead of stronger as you state? Or do you mean to say 'we also get a decrease in convective activity over the BoB ...'*

The negative regressions for MIDX onto OLR (indicated by the colours ranging from yellow to read) show an increase in convective activity over wide areas of Myanmar, the southern part of BoB, the Indian subcontinent, southern China, Hanai and Taiwan (at the eastern edge of this region). This indicates that the convective activity is increasing with increasing MIDX in large areas where the JA OLR is usually at its lowest values.
We included a more detailed description of the locations in the revised manuscript.

*Appendices: The Appendices are too terse to be useful. The autocorrelations discussed in Appendix B are not referred to in the main text (except at the end of Sec. 3 which merely states that autocorrelations were treated) and should be removed unless a more articulate discussion of how the autocorrelations affect the analysis is provided.*

Removed.

*Appendix A is also not necessary. You could simply mention that the criteria for significance are derived from the Z-test and refer the reader to Stouffer et al. (1949) and Whitlock (2005) for details.*

The Appendix A is removed from the manuscript.

*Fig. 1 caption: Add the term 'WIDX' to the description. For example, 'bottom: index for the Walker circulation (WIDX; solid)'*

The figure 1 is now redrawn and does no longer include the WIDX time series. It is replaced by a figure showing the time series of the JA basis functions MIDX, QBO, and ENSO, used in the regression analysis.

*P. 6 L. 4: Regarding 'graduate'. Do you intend this word to mean 'to make more gradual'? If so, this is an awkward use – if not, it's difficult to understand the meaning of the phrase. It would be better to use a different word.*

The term 'graduate' is a statistical/mathematical term to indicate the ability of the multi-model average to balance the extremes of individual CCMs. But we agree with the reviewer that it is not commonly used. We replaced the term 'graduate' by 'level out', that hopefully is better understood.

*P. 6, L. 8: Change 'indication for the' to 'indication of the'.*

Done.

*P. 6 L. 10: No comma after 'model'.*

Done.

*P. 7 L. 5: Change 'allows to express' to 'allows us to express' or some other grammatically correct wording.*

The sentence is rephrased.

*P. 7 L. 24: The clause 'whereas the nino3.4 ... variability' is a non sequitur – it implies contrasting behavior but no source of the contrast is given. Perhaps you intend to say something like 'the regression onto WIDX emphasizes the west Pacific circulation response to inter-annual SST variations whereas regression onto nino3.4 describes the larger (scale) response to ENSO variability.'*

It was intended to emphasize the ability of the nino3.4 index to better reflect the ENSO variability compared to the WIDX time series, as only the largest ENSO warm events are equally good captured by both indices. We are not talking about the regression of any of the indices in this section of the paper.
As Figure 1 is redrawn and does no longer include the WIDX, the discussion about the differences to the nino3.4 index is no longer included.

*P. 10 L3: Change relative to 'relatively'. Also you should state what you are comparing to when you say it is relatively small (i.e., relative to what?).*

We intended to express the smaller spread in JA velocity potential maxima of the CCMs over southeast Asia compared to the spread in stream function maxima within the AMA of the CCMs. Relative to the stream function maxima the velocity potential maxima deviate less about the multi-model average. We rephrased the sentence.

*P. 12 L. 6: Add (e.g., in parentheses) that the heating rates are displayed with red lines.*

Done.

*P. 21 L. 13: Change 'round' to 'around'*

We changed the term.

*P. 21 L. 25: Change 'South to the AMA' to 'South of the AMA'.*

Done.

*P. 22 L. 5: Remove 'in' from 'temperature from in nine re-analysis datasets'.*

Done.

[Figure]

**Figure 1.** Composited anomalies for wet (left) and dry (right) 360 K monthly mean ERA-Interim water vapor extrema in July and August over the TP region (30–40°N, 70–100°E) analysed for years from 1979–2013; ERA-Interim temperature (top) and water vapour (bottom). The ERA-Interim data are preprocessed and do not include QBO and ENSO variability. Overlaid as streamlines in grey are the composited horizontal wind anomalies; the $3513 \times 10^2$ m$^2$ s$^{-2}$ contour of the Montgomery streamfunction is overlaid in black.

**References**

Dethof, A., O'Neill, A., Slingo, J. M., and Schmit, H. G. J.: A mechanism for moistening the lower stratosphere involving the Asian summer monsoon, Q.J.R. Meteorol. Soc., 125, 1079–1106, 1999.

Ploeger, F., Günther, G., Konopka, P., Fueglistaler, S., Müller, R., Hoppe, C., Kunz, A., Spang, R., Grooß, J.-U., and Riese, M.: Horizontal water vapor transport in the lower stratosphere from subtropics to high latitudes during boreal summer, J. Geophys. Res.: Atmos., 118, 8111–8127, doi:10.1002/jgrd.50636, http://dx.doi.org/10.1002/jgrd.50636, 2013.

SPARC CCMVal: SPARC Report No 5 (2010) Chemistry-Climate Model Validation, WCRP-132, WMO/TD-No. 1526, 2010.

Tanaka, H. L., Ishizaki, N., and Kitoh, A.: Trend and interannual variability of Walker, monsoon and Hadley circulations defined by velocity potential in the upper troposphere, Tellus A, 56, 250–269, doi:10.3402/tellusa.v56i3.14410, 2004.

---

## Author Comment (AC3) · 13 May 2016

**Response to referee #3.**

We thank the anonymous referee for the constructive comments that helped to improve the paper. All further questions/comments are answered/annotated in the following, the manuscript is changed accordingly. (Referee comments are emphasized in *italics*.)

*General:*

*Validation of the chemistry climate models CCMs) with respect to their abilities to represent the Asian summer monsoon (ASM), especially the Asian monsoon anticyclone (AMA) is an important task for the atmospheric community. The paper uses*
10 *MIPAS and ERA-Interim data to validate such CCMs; the comprehensive analysis is clear and well presented. In the second part, the interannual variability of the ASM/AMA system is considered. However, there are some major points which need a more detailed discussion.*

*Major points:*

15

*1. Fig 3 and 4*
*Both figures show the results relative to the tropopause pressure that is certainly a good idea. You write that in order to account for differences among the CCMs in the location of AMA, the mean anomaly averaged over 30 degree was "centered where the 150 hPA eastward directed divergence free zonal wind maximizes". However, I would like to see such differences in the model*
20 *representation and would recommend to use a much simple averaging over 120-160E. Maybe you can make two figures for this (you do something similar in Fig 8). Furthermore, the most important information shown in Fig 3/4 are for me temperature anomalies (rather than wind anomalies) which are extremely difficult to read. A compromise could be to show wind anomalies in the absolute range 120-160E and temperature anomalies by using the relative coordinate defined by the wind maximum (and only to mention in the text that such "shifted" wind patterns are very similar for ERA-Interim and the MMOD analysis) .*

25

Including an additional Figure into the manuscript will further enlarge it. In view of the statement included in the second major point and also referring to anonymous referee 2, Fig. 3 and 4 are no longer contained in the manuscript but rather moved to the supplementary material, provided with the revision of the manuscript.
However, we followed the recommendation of the referee to present two kinds of sectional averages for the core region of
30 the AMA with one highlighting the temperature anomalies (shaded) for a fixed range 60-120°E (Fig. S2) and one with the currently used method to create the sectional averages and highlighting the wind anomalies (shaded) (Fig. S1).
The motivation for the original Fig. 4 (now Fig. S3) is to show the different meridional velocities at the western and eastern edges of the AMA, therefore the shading should highlight these differences.

35 *2. Fig 10 and the discrepancy with Randel et al 2015*
*This is a very interesting and important point. However, a simple explanation referring to "different approach" is not enough for me. You can certainly repeat the Randel's procedure by using ERA-Interim H2O (instead of MLS like in Randel et al.). If you get a similar picture ("more convection makes a dry anomaly") than is your statement ("different approach") correct. Otherwise, without such a test you have a "confusing result" if compared with the published work of Randel et al 2015.*

40

We followed the advice of the anonymous referee and analysed the ERA-Interim data (water vapour, temperature, and horizontal wind components) and NOAA OLR data in the same way as it was done by Randel et al. (2015) (R15). We created wet and dry composites on the basis of daily data from May to September for years from 2005 – 2013, to be as much comparable to R15 as possible (Fig. 1). We did the same kind of analyses for daily data from July to August for years from 1979 – 2013
45 (Fig. 2), to be comparable with the regression analyses of the manuscript. As in R15 the ERA-Interim data were pre-processed with the multiple linear regression model (MLR), to create a time series of daily ERA-Interim data without trend, and QBO/ ENSO induced variability. When using the same time periods as R15, i.e. data from May to September for years from 2005 – 2013, we get similar OLR anomalies over the region 20-30°N/90-120°E, which was identified by R15 as the key convective region, showing reduced convective activity over this region for the wet composite and intensified convective activity for the

dry composite (s. Fig. 1, top). Also the structure of the temperature anomalies are similar to R15, although less pronounced for the wet anomalies (s. Fig. 1, second row).

When we repeat the analyses with data covering only July to August but using the years from 1979 – 2013 we can confirm the OLR anomalies shown in Figure 1 over the region 20-30°N/90-120°E and also the structure of the temperature anomalies, with the temperature anomalies of the wet composite now more pronounced. But for the adjacent region in the south, extending from the Indian subcontinent, the Bay of Bengal (BoB) to Vietnam, we get the opposite anomalies with more intense convective activity for the wet composite and reduced convective activity for the dry composite (s. Fig. 2, top). Based on the results of Figure 2 and based on the results of the MLR for the monsoon circulation index (MIDX) (see Figure 10 of the manuscript) we come to the conclusion that the wet anomalies are connected with a more intense Asian summer monsoon. From this comparison it seems that the differences to R15 can partly be explained by a different time period used in their study. It seems that the differences to R15 are partly due to a different representation of the ASM intensity. The conclusions drawn from the additional analyses are included in the revised manuscript. Two additional Figures are part of the supplementary material.

*. . . Furthermore, the paper is in my opinion too long. I would recommend to publish two parts: (1) validation with MLS/ERA-Interim and (2) Iterannual variability. But, that is your decision.*

Section 3 is now shortened, as Fig. 3 and 4 moved to the supplementary material. However, we don't want to separate the paper into two parts.

*Minor points:*

*1. General*
*In almost all your figures you use a matrix of sub-panels. It would be easier to read such figures if you would denote every row and every column separately. E.g. Fig 5/6 θ = 360 370, 380 K for the rows and MIPAS/MMOD for the columns.*

We followed the advice and organised the labelled Figures by rows and columns.

*2. P1, abstract, L14-15*
*please mention "zonally asymmetric ENSO response versus zonally symmetric QBO modulation"*

We have included a more detailed description of the QBO modulation in the abstract of the revised manuscript.

*3. P1, L24*
*first "wave-driven" forcing, followed by heat transport from the tropics to the high latitudes and, finally slow ascent due to radiative heating - please reformulate*

The intention was to characterize the vertical transport confined to tropical latitudes, rather than to describe the full picture of the meridional tracer transport in the middle atmosphere. The section is now rewritten to avoid any misunderstanding.

*4. P 2, L 20-25*
*To discuss the importance of the Tibetan Plateau you should also mention the Boos and Kuang, Nature 2010 paper stating that for the formation of the Asian monsoon circulation pattern orography is the most important factor and the impact of sensible heat (Tibetan Plateau) is rather a second order effect*

This alternative result of Boos and Kuang (2010) are now cited in the revised manuscript.

*5. P3 L 10-13*
*Maybe you should discus it more carefully: the core of the anticyclone is rather in the extratropics than in the tropics. Furthermore, the anticyclone itself acts more as an isentropic blower. Inside of the anticyclone the the tropospheric pollution are*

*trapped and probably transported into the TTL (Randel et al., Science, 2010). Outside of the anticyclone a strong in-mixing of stratospheric signatures into the TTL happens (see related paper from Konopka et al and Ploeger et al)*

This paragraph is now revised and extended by a short comment on in-mixing, as it was analysed by Konopka et al. (2009) and Ploeger et al. (2012).

*6. P3 L14*
*...(QBO) or the "internal variability of the ASM iteslf".*

This is now specified as proposed.

*7. P4, L26*
*"aspects of the climatological state are compared with" - which aspects, please reformulate*

This is now formulated more specifically.

*8. P6, caption Fig 1*
*please use the abbreviation WIDX*

The revised version of Figure 1 does not include the Walker circulation index (WIDX) any more.

*9. P6, L5*
*"graduate" - I am not sure that this is a right word. Maybe "mask" or "suppress"*

We reformulated the sentence.

*10. P7, L5*
*Explain the vector k*

Done.

*11. P9 L2*
*Use the notation $\psi$ for the divergence-free part of the flow. Same for $\chi$ (which were defined in the previous section).*

Done.

*12. P12 Fig 5*
*The enhanced signatures of H2O north of 30N seem to propagate eastward mainly by planetary waves as described by Ploeger et al. Maybe you would like to include some comments about this point*

A short statement on this is now included in the manuscript and the reference to Ploeger et al. (2013) is given.

*13. P14 Fig 7*
*There are much lower temperatures at 380 K for MMOD than for ERA. On the other side MMOD are moister compared with MIPAS. You should comment this point*

There are different points used for the statistics of the H2O maximum and the temperature minimum in the region enclosed by the rectangle at 380 K. Whereas the lowest temperatures are located at the southern edge of the AMA the maximum in H2O

within the rectangle is located on average at the north-eastern corner of the rectangle.

*14. P14 L9*
*"O3 in the UTLS can better serve as a passive tracer..." - maybe you can make this point earlier, e.g. as you introduce O3 into your discussion*

The sentence has moved to the beginning of the paragraph.

*15. P15 Fig 8*
*After the major point 1 was included, Fig 8 would be easier to understand*

The major point 1 is only included in the supplementary material, but we hope that the information given by the additional Figure can be helpful to understand the Figure still included.

*16. P15 last sentence and P16 first sentence*
*This feature was discussed in literature as in-mixing, see Konopka et al 2009, 2010, Ploeger et al 2012. Maybe you would like to include these references into your discussion*

The suggested references are now taken into account.

*17. P17 caption of Fig 7 (and Fig 2)*
*You introduced the decomposition given by the eq (1) but you do not use the introduced notation. Please state it explicitly if you show $\psi$, $\chi$, $\chi^*$, etc.*

Done.

*18. P17 L13*
*For me MIDX is a more direct measure of the anticyclone rather than of the whole ASM system*

Please see our response to point 21 below.

*19. P18 Fig 10 and the discrepancy with Randel et al 2015 see major point 2*

Please see our response to major point 2.

*20. P19 Fig 11*
*I think, you use the ERA-Interim related results too strong as a benchmark for the following investigations. Whereas ERA-Interim temperatures and probably H2O are good enough for your study, ERA-Interim ozone around and below the tropical tropopause is probably not good enough for that (mainly because only O3 column is constrained by sat elite observations as described in Dragoni et al., 2011) . In the following you describe large differences in ozone between multimodel average of the CCMs and the ERA-Interim. I would recommend to exclude completely the ERA-Interim ozone.*

Because ozone has a relatively long lifetime in the UTLS region it can be used as a complementary dynamical tracer. Due to a lack of a direct observational constraints we should not expect that ERA-Interim ozone matches absolute (in-situ observed) ozone values well, but interannual variability and thus regression patterns should generally be matched reasonably well. We have included a "health warning" in the manuscript.

*21. P20 L7*
*"As MIDX is a direct measure of the strength in upwelling" - for me MIDX is a direct measure of the (divergence) of the*

*anticyclone, please re-formulate*

The MIDX is created from the transient eddy part (i.e. after subtracting the zonal average) of the velocity potential ($\chi^*$'). The maxima in $\chi^*$', the individual values of the MIDX, are thus taken from a purely divergent field. We agree with the referee that the horizontal divergence is connected to the horizontal, purely rotational (divergence free) circulation, as the maximum of the horizontal divergence occurs where the streamlines of two reversed circulation cells diverge. On the western side of the respective maximum in $\chi^*$' the anticyclonic flow of the AMA is located, whereas on the eastern side cyclonic flow prevails. For continuity reasons the divergence has to be connected with upwelling, which can be seen as connected with convective uplift.

We have reformulated the sentence to emphasize the importance of the upper tropospheric divergence.

*22. P20 L9*
*...or have increased H2O or less O3*

*23. P20 L10*
*...or decreased H2O or higher O3*

This serves as a general introduction of the MIDX regression coefficients for positive or negative signs. When a variable increases with increasing MIDX, we get a positive regression coefficient and vice versa. The statements are not meant to describe the regression coefficients that should be expected in the monsoon anticyclone.

*24. P21 L17*
*"The negative O3 caused...." - I do not understand your explanation. Negative O3 anomaly means a stronger tropospheric influence (more upwelling) that is in agreement with the positive H2O anomaly. Please clarify*

Both, the negative O3 anomalies and the negative temperature anomalies are nearly co-located with each other and also with the anticyclonic anomalous circulation pattern. There is no contradiction to the positive H2O anomaly that extends over larger parts of the AMA (at 380 K as well as at 100 hPa). The positive H2O anomaly seems to be more related to the positive temperature anomaly at the southern edge of the AMA. This is now also discussed in the revised manuscript.

*25. P22 L15*
*"unexpected positive response" - see comments above to ERA-Interim ozone*

We added a comment about the limitations of the ERA-Interim O3 data.

*26. P25 L23*
*"many regions" - please list these regions*

The regions are now explicitly listed.

*27. P26 L17*
*...suggest transport of H2O through this region*

H2O is now explicitly stated.

[Figure]

**Figure 1.** Composited anomalies for wet (left) and dry (right) 100 hPa ERA-Interim water vapor extrema from May to September over the Asian monsoon region (20–40°N, 40–140°E) analysed for years from 2005–2013, from top to bottom for NOAA OLR, ERA-Interim temperature, water vapour, and ozone without QBO and ENSO variability. Overlaid as streamlines in grey are the composited horizontal wind anomalies; the 16.750 m geopotential height contour is overlaid in black. Results for OLR are shown averaged 0–10 days prior to the stratospheric water vapour extrema; overlaid red contours indicate climatological OLR values $\leq 220$ W m$^{-2}$. Adapted from Randel et al. (2015).

[Figure]

**Figure 2.** As Fig. 1 but using data from July to August for years from 1979–2013.

**References**

Boos, W. R. and Kuang, Z.: Dominant control of the South Asian monsoon by orographic insulation versus plateau heating, Nature, 463, 218–222, doi:10.1038/nature08707, http://dx.doi.org/10.1038/nature08707, 2010.

5    Konopka, P., Grooß, J.-U., Plöger, F., and Müller, R.: Annual cycle of horizontal in-mixing into the lower tropical stratosphere, J. Geophys. Res.: Atmos., 114, doi:10.1029/2009JD011955, http://dx.doi.org/10.1029/2009JD011955, d19111, 2009.

Ploeger, F., Konopka, P., Müller, R., Fueglistaler, S., Schmidt, T., Manners, J. C., Grooß, J.-U., Günther, G., Forster, P. M., and Riese, M.: Horizontal transport affecting trace gas seasonality in the Tropical Tropopause Layer (TTL), J. Geophys. Res.: Atmos., 117, doi:10.1029/2011JD017267, http://dx.doi.org/10.1029/2011JD017267, d09303, 2012.

Ploeger, F., Günther, G., Konopka, P., Fueglistaler, S., Müller, R., Hoppe, C., Kunz, A., Spang, R., Grooß, J.-U., and Riese, M.: Horizontal

10    water vapor transport in the lower stratosphere from subtropics to high latitudes during boreal summer, J. Geophys. Res.: Atmos., 118, 8111–8127, doi:10.1002/jgrd.50636, http://dx.doi.org/10.1002/jgrd.50636, 2013.

Randel, W. J., Zhang, K., and Fu, R.: What controls stratospheric water vapor in the NH summer monsoon regions?, J. Geophys. Res.: Atmos., doi:10.1002/2015JD023622, 2015.

---

## Author Response (AR2)

Berlin, 24[th] June 2016

Dear Dr. Jianzhong Ma

We submit the second revised version of our manuscript (doi: 10.5194/acp-2015-991):

**"Interannual variability of the boreal summer tropical UTLS in observations and CCMVal-2 simulations"**

**by Markus Kunze, Peter Braesicke, Ulrike Langematz, and Gabriele Stiller**

We have considered all comments and suggestions for modifications by the reviewer and have also considered your editorial comments. A detailed reply to the reviewer is attached to this letter. Furthermore we have highlighted all changes in the manuscript and also attached them to this letter.

We have again shortened the manuscript by removing Figure 6 (now included in the supplementary material) and related discussion from Section 4. We also removed a short paragraph on page 14, as requested by the reviewer.

We are confident that we could address sufficiently the comments of the reviewer and hope that the second revised version will now be acceptable for publication.

Yours sincerely,

Dr. Markus Kunze (on behalf of all co-authors)

**Response to referee #2.**

We thank the referee again for the constructive criticism. All further questions/comments are answered/annotated in the following, the manuscript has been changed accordingly (referee comments are emphasized in italics).

*General comments:*
==========================
*It is clear that the authors put considerable effort into the revision. While the paper remains an unsatisfying mix of qualitative comparisons and speculation regarding dynamical connections, with Sec. 5 being particularly unfocused and having too wide of a scope, there is enough interesting and new material to warrant publication. Below are suggestions for minor alterations.*

*Specific Comments:*
================================

*Page 2, line 3: By saying 'contrasts with', do you mean 'is balanced by'?*

The small net radiative heating, thus the radiative imbalance, is due to the SW heating being larger than the LW cooling. Ascending motion in the TTL leads to an adiabatic cooling. Decreases in TTL temperatures lead to decreased outgoing LW radiation that largely balances the absorbed SW radiation, still resulting in a small net heating rate. The diabatic ascent in the TTL and the associated adiabatic cooling thereby avoids a relaxation towards radiative equilibrium. We changed 'contrasts' with 'balanced'.

*P. 2 L. 8: With regards to ice injected into the lower stratosphere; isn't evaporation due to mixing with sub-saturated air a more likely moistening process than sublimation?*

Sublimation is defined as the direct transition from the solid phase into the gas phase  However, we agree that evaporation might be understood as a more generic term by many readers (see e.g. Corti et al., 2008) and we have changed the wording accordingly.

*P. 2 L. 9: Change 'emphasized to contribute' to 'emphasized as a contributor' or, maybe, 'recognized as a contributor'.*

Done.

*P. 6 L.6 (also discussion of Figs. 2-7): Regarding using extremes for comparing models: I would expect there to be (in general) more model-to-model variability with extremes (such as maximum velocity potential within a box) than with average quantities. To put the inter-model comparisons into perspective, it would be good to mention comparisons of average values as well.*

The bar-charts on the right side of the figures showing multi-model averages do also represent the long-term averages of the respective CCMs. The wording 'maximum' is referring to the fact that the largest value of the respective variable within a certain region has been taken. Therefore this maximum is also a long-term average and not the extreme July/August value of the whole simulation.

*P. 7, L. 15 and 27: The use of 'time-mean' to described the second term on the RHS of Eq. 1 is confusing because you also say that is describes a time-varying Walker circulation. Perhaps it would be better to separate the two eddy terms based on time-scale. The second term of Eq. 1 being a slowly varying (inter-annual) quantity and the last term being a more rapidly varying (mostly seasonal) quantity.*

We are referring to Tanaka et al. (2004) who introduced the method in the way it is described in the paragraph. The second term is at first introduced as the time independent, stationary term to motivate the transient term. It is clearly described that for the second term the time variation is included by using a moving average. We therefore do not see any need for further clarification.

*P. 8 L. 8-9: What is meant by 'the two innermost tropical latitudes? Please be more specific; e.g., specify a latitude range over which values are averaged.*

The latitudes used are now explicitly given.

*P. 8: Regarding Eq. 2: Please describe the performance of this regression. In particular: How independent are each of the terms? And how much of the low-frequency variance is unaccounted by the primary terms of the regression (or, equivalently how much of the low-frequency variance is accounted for by the residual)?*

The explained variance (like the regression coefficients) varies in all three dimensions. When using the adjusted coefficient of determination (adjR[2)]) as a measure of the overall performance of the multiple linear regression model, large regions of an individual isentropic level have an adjR$^2$ larger than 0.9. As an estimate of the variance accounted for by the residual, we calculated the quantity $(1 - \text{adj}R^2)$, implying that in large regions less than 10% of the total variance is not accounted for by the MLR basis functions and therefore described by the residuals. This overall result is of cause somewhat different between the variables and also varies between the CCMs. Comparison of the MLR performance for ERA-Interim with the results for individual CCMs shows a better performance of the MLR for ERA-Interim. Please see details below.

*Later in the manuscript, the authors discuss the statistical significance of their regressions – however, statistical significance does not necessarily indicate a meaningful regression relationship since one can render any relationship statistically significant simply by using more data. That is, one quantity might only explain 1% of the variance of another, yet with enough data, that 1% can be statistically significant.*

In the most problematic case (some small region over the Tibetan Plateau, visible in water vapour at 360K) the residual (unexplained) variance can be 30% (in all other heights and regions the unexplained variance is much smaller), thus 70% of the overall variance are still explained by the regression model. From this 70% a smaller fraction is explained by MIDX, larger fractions by QBO and NINO3.4. We do not see a problem with a significant result that explains locally sometimes a small fraction of variance, as long as it is embedded in a larger region that is physically consistent and where larger fractions of variance are explained as well.

We added a paragraph to section 3.3.2 with a statement regarding the performance of the MLR.

*P. 10 L. 13: It is not clear what is meant by 'the onset of the divergent winds'. For example, do you mean that the location coincides with where winds become divergent (i.e., switch from convergent to divergent) or that the location coincides with the maximum values of divergence, or maybe some other meaning?*

We meant to highlight the fact that the divergent winds blow out of the regions of the maximum values in velocity potential. As the intensity of the divergent wind is proportional to the gradient of velocity potential, the center of the maxima can be regarded as a starting point of purely divergent wind vectors.
We have rephrased the sentence.

*P. 10 L. 13: 'vice-versa' seems to be misused here. Do you mean that the negative peaks values of chi correspond to convergence? If that is the case consider rewriting to something like 'The positive peaks ... of upwelling, coincide with divergent winds while the negative peaks coincide with convergence.'*

Done. Please see also the answer to the previous point.

*P. 10 last paragraph: You seem to be stating that the large values of humidity in the northern extra tropics seen in the multi-model-mean (Fig. 3) is due to lateral transport from the tropics. If that is true, then the fact that the values of humidity in the extra-topics is as large as values within the anti-cyclone implies that a large fraction of extra-tropical air was recently transported northward by the anti-cyclone (if not then air from the anti-cyclone would be mixing with air that is, presumably, drier, which would reduce the humidity). That implication is difficult to believe.*

Lateral transport related to the ASM is present but the MMOD H2O mixing ratios of the CCMs are also relatively high at high northern latitudes, as some CCMs, especially EMAC-FUB and SOCOL, are overestimating the H2O mixing ratios in the lowermost stratosphere, probably due to their coarse vertical resolution near the tropopause.
The revised manuscript includes a sentence about this.

*P. 10 L. 3: Change 'diabatic' to 'radiative'.*

Done.

*P. 12 L. 3-6: This paragraph discussing temperature has no obvious bearing on the paper and the reasoning seems weak. Either clarify or remove.*

A clarifying sentence is added to this section.

*P. 12 L. 7-8: Remove the phrase 'and as it is not affected by dehydration'*

Done.

*P. 14 L. 1-4: Are you trying to say that ozone values over Asia exhibit variations that coincide with variations of the anti-cyclone on time scales less than the seasonal values you show? Please revise this paragraph to make its meaning clear or remove it since your study does not examine sub-seasonal variability.*

The paragraph has been removed from the manuscript.

*P. 14 L. 5 – P. 15 L. 6: The paragraph that discusses Fig. 6 is very weak. I suggest removing it along with Fig. 6.*

Although we find it still interesting to show the latitude-height sections of the water vapour and temperature anomalies related to the ASM circulation, we followed the suggestion of the referee and removed Figure 6 and the related discussion from the manuscript. Figure 6 is now part of the supplement.

*P. 19 L. 1: Vice versa is misused. Revise to something like '… dipole structure with warmer conditions on the southern edge of the AMA during wet phases with the oppositely signed structure evident in the dry phases'. If you want to use 'vice versa' you could say 'the temperature structure has a warm anomaly in the south and a cold anomaly in the north during the wet phase and vice versa during the dry phase.' [although I would not use vice versa this way either.]*

The sentence is now rephrased.

*Last paragraph of P. 19 and first paragraph of P. 20; Fig. 8: Don't you find it interesting that the strongest wet anomaly is nearly collocated with a strong cold anomaly?*

This is indeed interesting. The cold anomalies are also collocated with the anticyclonic circulation anomalies. Park et al. (2007) describe the lower temperatures near the tropopause (i.e. near 100 hPa) as a result of the large-scale balanced dynamics and not the result of convective overshooting. They also describe the decoupling of the water vapour maximum within the AMA near the tropopause from the region of strongest convection. A sentence about this is added to the discussion of Figure 9a, showing the results on the 380 K isentropic level.

*Technical details:*
==================================
*P. 6 L. 5: Remove comma after 'CCMs'*

*P. 8 L. 8: Change 'wind in 50 hPa' to 'wind at 50 hPa'.*

*P. 12 L.3: Either use 'The temperatures on … are influenced' or*
*'The temperature on … is influenced'.*

*P. 12 L. 8: Change 'ASM are the' to ASM is the'.*

All technical corrections are done.

[revised manuscript text omitted]